# Structural basis of Cullin 2 RING E3 ligase regulation by the COP9 signalosome

Sarah V. Faull[1,5], Andy M.C. Lau [2,5], Chloe Martens[2], Zainab Ahdash[2], Kjetil Hansen [2], Hugo Yebenes[1,3], Carla Schmidt[4], Fabienne Beuron[1], Nora B. Cronin[1], Edward P. Morris [1] & Argyris Politis [2]

Cullin-Ring E3 Ligases (CRLs) regulate a multitude of cellular pathways through specific substrate receptors. The COP9 signalosome (CSN) deactivates CRLs by removing NEDD8 from activated Cullins. Here we present structures of the neddylated and deneddylated CSN-CRL2 complexes by combining single-particle cryo-electron microscopy (cryo-EM) with chemical cross-linking mass spectrometry (XL-MS). These structures suggest a conserved mechanism of CSN activation, consisting of conformational clamping of the CRL2 substrate by CSN2/CSN4, release of the catalytic CSN5/CSN6 heterodimer and finally activation of the CSN5 deneddylation machinery. Using hydrogen-deuterium exchange (HDX)-MS we show that CRL2 activates CSN5/CSN6 in a neddylation-independent manner. The presence of NEDD8 is required to activate the CSN5 active site. Overall, by synergising cryo-EM with MS, we identify sensory regions of the CSN that mediate its stepwise activation and provide a framework for understanding the regulatory mechanism of other Cullin family members.

---

[1] Division of Structural Biology, The Institute of Cancer Research, London SW3 6JB, UK. [2] Department of Chemistry, King's College London, 7 Trinity Street, London SE1 1DB, UK. [3] Centro de Investigaciones Biológicas, Consejo Superior de Investigaciones Científicas, Madrid, Spain. [4] Interdisciplinary Research Center HALOmem, Charles Tanford Protein Centre, Martin Luther University Halle-Wittenberg, Kurt-Mothes-Strasse 3a, 06120 Halle/Saale, Germany. [5] These authors contributed equally: Sarah V. Faull, Andy. M. C. Lau. Correspondence and requests for materials should be addressed to E.P.M. (email: ed.morris@icr.ac.uk) or to A.P. (email: argyris.politis@kcl.ac.uk)

Cullin-RING Ligases (CRLs) are modular, multi-subunit complexes that constitute a major class of ubiquitin E3 ligases[1,2]. CRLs coordinate the ubiquitination of substrates as either a signal for degradation via the 26S proteasome, or to alter the function of the target protein[2,3]. The CRL2 E3 ligase consists of a Cullin 2 (CUL2) scaffold in association with a catalytic RING-box protein (RBX1), with the substrate adaptors Elongin B (ELOB) and C (ELOC) at its N-terminal[4]. When associated with the von Hippel–Lindau (VHL) tumour suppressor substrate receptor, the CRL2 complex is the primary regulator of the Hypoxia Inducible Factor 1-α (HIF-1α) transcription factor[5,6]. Mutations in the interface between VHL, ELOB and ELOC can deactivate CRL2 leading to an accumulation of HIF-1α, which can in turn drive tumorigenesis through the over-activation of oncogenes[7]. Moreover, CRL2 has recently been identified as a potential target for small molecular inhibitors and PROTACs—a new class of cancer drugs that promote degradation of tumorigenic gene products[8–10]. These fascinating systems have been described in detail by a number of excellent reviews[1–3].

Activation of CRL2, in common with other members of the CRL family, involves a cascade of E1, E2 and E3 enzymes, which conjugate the ubiquitin-like protein NEDD8 (N8) to residue K689 on the CUL2 scaffold[11]. In its activated state, CRL2~N8 (the ~ stylisation denotes a covalent interaction) recruits the ubiquitin-conjugated E2 enzyme via the RING domain of RBX1[12]. Ubiquitination now takes place, covalently adding ubiquitin to the substrate molecule docked at the CRL2 N-terminal. The activity of CRL2 is negatively regulated by the 331 kDa Constitutive Photomorphogenesis 9 Signalosome (CSN) complex, frequently referred to as the COP9 signalosome complex[13–15]. The CSN was originally identified as consisting of eight subunits (designated as CSN1–8 by decreasing molecular weights of 57–22 kDa), and is organised in a splayed hand architecture, which has high sequence and structural homology to the proteasome lid[13,14,16–18]. CSN1, 2, 3, 4, 7 and 8 are structurally homologous to each other and together contribute to the fingers of the splayed hand which arise from extended N-terminal α-helical repeats[16,18]. Each CSN1-8 subunit includes an extended C-terminal helix which associates together forming a C-terminal helical bundle. CSN5 and 6 are also closely related structurally and form a globular heterodimer located on the palm of the hand. CSN5 is responsible for the deneddylase activity of the CSN. A ninth subunit, CSNAP, has recently been identified, and is thought to play a role in stabilising the CSN complex[18].

Electron microscopy (EM) based structural analysis has provided important insights into the mechanism of CRL1 regulation by CSN[16,19]. CRL4A and CRL3 have also been observed to form such complexes[20]. However, despite intense interest, structural information of the CSN bound to CRLs remains limited to CRL1[12,16,19], CRL4A[20], and a low-resolution map of a dimeric CSN–CRL3~N8[20] complex. In particular, the analysis of the CSN–CRL4A~N8 complex[20] identified at least three major steps by which CRL~N8 is deneddylated by the CSN. In the first step, the extended N-terminal helical modules of CSN2 and CSN4 conformationally clamp the C-terminal domain of the CRL4A~N8 and RBX1[16,19,20]. The second step involves the release and consequent relocation of CSN5/CSN6 closer to NEDD8, brought about by disruption of the CSN4/CSN6 interface[20]. Disrupting the binding interface between CSN4/CSN6 through removal of the CSN6 insertion-2 loop (Ins-2), resulted in enhanced deneddylase activity[13], presumably due to more complete release of CSN5/CSN6. In the final step, the mobile CSN5 binds to NEDD8, leading to deneddylation via its JAB1/MPN/MOV34 (JAMM) metalloprotease domain[21]. The JAMM motif consists of H138, H140 and D151 zinc-coordinating residues, and

residue E104 of the CSN5 insertion-1 loop (Ins-1)[13]. In apo-CSN, the Ins-1 loop occludes the CSN5 active site, auto-inhibiting the deneddylase[13,22]. Deneddylation is also severely diminished by a H138A point mutation in CSN5[13].

Surprisingly, the CSN can also form complexes with each of the Cullin 1–5 family members even without NEDD8[23]. Free CRLs such as CRL1 have been reported to readily bind and inhibit the CSN, albeit at relatively lower affinity than the neddylated CRL1[24]. While the exact role of CSN–CRL complexes remains unclear, it has been hypothesised that these complexes may function to regulate the cellular level of ubiquitin ligase activity of CRLs once they have been deneddylated, effectively sequestering E3 ligases from the intracellular environment[24].

Building on the existing knowledge of the CSN–CRL systems, here we pose the question: are similar structural changes to be found in other CSN–CRL complexes, and how does binding of neddylated CRLs lead to activation of the CSN5 catalytic site? To address this, we present structures of the CSN–CRL2~N8 complex, together with the structure of the CSN–CRL2 deneddylation product. We complement our cryo-EM analysis with chemical cross-linking mass spectrometry (XL-MS) allowing us to clarify the positions of particularly dynamic regions in the complexes. We use hydrogen-deuterium exchange mass spectrometry (HDX-MS) to interrogate the role of the CSN4/CSN6 interface in communicating CRL binding to the CSN5 active site. Overall, our structures of the CSN–CRL2~N8 and its deneddylation product, the CSN–CRL2, reveal the intricate conformational changes of CSN that lead to deneddylation of the CRL2.

## Results

**Cryo-EM structures of the CSN–CRL2~N8 complex.** To study the molecular interactions between neddylated CRL2 (CRL2~N8) and the CSN, we performed single-particle cryo-EM to resolve a structure of the assembled CSN–CRL2~N8 (referred to as the holocomplex) (Supplementary Figs. 1 and 2). The H138A mutation in the catalytic site of CSN5 subunit makes it possible to assemble CSN–CRL2~N8 complexes in which NEDD8 remains covalently attached over the time scale of the experiment[16,19]. To justify the use of the H138A point mutation, we performed a band-shift assay comparing the deneddylation activity of the $CSN^{WT}$ and $CSN^{5H138A}$ mutant enzymes (Supplementary Fig. 3). As expected, $CSN^{WT}$ rapidly cleaves NEDD8 from CRL2~N8 within seconds of incubation, while little to no deneddylation activity is seen from the $CSN^{5H138A}$ complex (Supplementary Fig. 3). This mutant form of CSN was used throughout the work described below, unless otherwise specified.

Using 3D classification, we were able to generate maps of three different structures: (a) a holocomplex map at 8.2 Å, (b) a map of the complex with little or no density for VHL at 8.0 Å, and (c) a map of the complex with little or no density for CSN5/CSN6/VHL at 6.5 Å (Supplementary Figs. 2, 4, and 5). The two partial complexes likely arise from compositional heterogeneity in the original samples from which the structural analysis has succeeded in isolating subpopulations. To verify the existence of subcomplexes, we subjected the CSN–CRL2~N8 to native MS (Supplementary Fig. 6, Supplementary Data 1). In line with subpopulation observations from our cryo-EM, we identified subcomplexes of the CSN–CRL2~N8 missing the CSN5, VHL, ELOB or ELOC subunits. These observations support the notion that similar levels of heterogeneity observed in the cryo-EM analysis of other CSN–CRL complexes, CSN–CRL1~N8 and CSN–CRL4A~N8[19,20] is also likely to arise from variable subunit composition and may be ubiquitous to all CSN–CRL complexes.

Next, we fitted into each map, the highest resolution crystallographic structure of the CSN (PDB 4D10)[13] and a homology

**Table 1 Cryo-EM data collection, refinement and validation statistics**

| | CSN–CRL2-N8 (holocomplex) (EMB-4739) (PDB 6R7F) | CSN–CSN5/CSN6-CRL2–VHL-N8 (EMB-4744) (PDB 6R7N) | CSN–CRL2–VHL-N8 (CSN5/CSN6 refined) (EMB-4742) (PDB 6R7I) | CSN–CRL2-N8 (VHL refined) (EMB-4736) (PDB 6R6H) | CSN–CRL2 (EMB-4741) (PDB 6R7H) |
|---|---|---|---|---|---|
| **Data collection and processing** | | | | | |
| Magnification | 47,170 | 47,170 | 47,170 | 47,170 | 47,755 |
| Voltage (kV) | 300 | 300 | 300 | 300 | 300 |
| Electron exposure (e-/Å$^{-2}$) | 45 | 45 | 45 | 45 | 83 |
| Defocus range (μM) | 1.8–3.0 | 1.8–3.0 | 1.8–3.0 | 1.8–3.0 | 1.8–3.0 |
| Pixel size (Å) | 1.060 | 1.060 | 1.060 | 1.060 | 1.047 |
| Symmetry imposed | C1 | C1 | C1 | C1 | C1 |
| Initial number of particles | 316,921 | 316,921 | 316,921 | 316,921 | 308,936 |
| Final number of particles | 20,055 | 22,471 | 24,552 | 24,049 | 17,191 |
| Map resolution (Å) | 8.2 | 6.5 | 8.0 | 8.4 | 8.8 |
| FSC threshold | 0.143 | 0.143 | 0.143 | 0.143 | 0.143 |
| Map resolution range (Å) | 6.7–22.7 | 5.9–8.0 | 5.9–11.5 | 6.3–17.4 | 5.6–16.8 |
| **Refinement** | | | | | |
| Subunits missing | None | CSN5/CSN6, VHL | VHL | None | None |
| Initial PDB used | 4D10, 5N4W, 4WQO, 3DQV | 4D10, 5N4W, 4WQO | 4D10, 5N4W, 4WQO | 4D10, 5N4W, 4WQO | 4D10, 5N4W, 4WQO |
| Number of residues | 3727 | 2974 | 3578 | 3716 | 3251 |

model of the CRL2 (including the VHL-ELOB-ELOC) using molecular dynamics flexible fitting (MDFF)[25] (Table 1; "Methods" section). In the model of the holocomplex (8.2 Å), the main interactions occur between the C-terminal end of CUL2 and the extended N-terminal helical repeats of CSN2 and CSN4 (Fig. 1a, b). Compared with their conformation in the highest resolution apo-CSN crystal structure[13] (PDB 4D10), CSN2 and CSN4 are moved by 30 and 51 Å, respectively, towards CUL2 (Fig. 1e, Supplementary Movie 1). We additionally reviewed the conformations of CSN2 and CSN4 in all nine available intact structures of the apo-CSN (PDBs 4D10, 4D18 and 4WSN; Supplementary Fig. 7). We compared each of the apo-CSN structures with our holocomplexes and with other fitted models of the CSN in complex with CRL1~N8 and CRL4A~N8. Significant structural variation in CSN2 and CSN4 is observed within these apo-CSN structures, but in all cases, their conformations were substantially different to any found in the holocomplexes. In each of our CSN–CRL2 structures, the clamping motion of CSN2 and CSN4 is a swinging rotation about hinges located close to the CSN2 and CSN4 winged helix domains. For CSN2 this is coupled with an additional rotation about the axis of the superhelix formed from its N-terminal helical repeats. In the case of CSN4 the movement is coupled to the detachment of CSN4 from the Ins-2 loop of CSN6 by ~30 Å and leads to an ~12 Å shift in CSN5 (Fig. 1c, f). Only minor conformational changes were found in the CSN1, CSN3, CSN7B or CSN8 subunits. In the CRL2~N8 moiety, a number of relatively small rearrangements of CUL2, RBX1, ELOB, ELOC and VHL subunits were observed compared to its crystal structure[26] (Supplementary Fig. 8a, b).

In our EM maps and the other published CSN–CRL structures[16,19,20], the exact position of NEDD8 and the CUL2 Winged-Helix B (WHB) domain were difficult to determine. To address this limitation, we carried out XL-MS experiments on the CSN–CRL2~N8 complex using the bis(sulfosuccinimidyl)suberate (BS3) cross-linker which targets lysine side chains ("Methods" section). We identified a total of 24 inter- and 60 intra-protein cross-links (Supplementary Data 2, Supplementary Fig. 9a, b). To generate a model of the CSN–CRL2~N8, we performed cross-link guided modelling which allows the placement of the WHB, NEDD8 and VHL subunits using identified cross-links from XL-MS ("Methods" section). We imposed a cross-link distance threshold of 35 Å which takes into account the length of two lysine side chains (15 Å), the BS3 cross-linker length (10 Å) and an extra 10 Å to allow for domain-level flexibility ("Methods" section). Our model of the CSN–CRL2~N8 satisfies all cross-link distances (Supplementary Fig. 9c). Three cross-links between CUL2-WHB (CUL2$^{K382}$-WHB$^{K720}$, CUL2$^{K382}$-WHB$^{K677}$ and CUL2$^{K433}$-WHB$^{K677}$) were used for the positioning of the WHB domain (Supplementary Fig. 9d, red text). A further two cross-links between CUL2 and NEDD8 (CUL2$^{K382}$-N8$^{K33}$ and CUL2$^{K433}$-N8$^{K6}$) allowed the positioning of NEDD8 near CSN5 (Fig. 1d, Supplementary Fig. 9d, green text). In this conformation, the isopeptide bond of NEDD8 is juxtaposed to the CSN5 active site. For the isopeptide bond of NEDD8 to reach the CSN5 active site, the WHB domain must be extended from its crystallographic conformation towards the CSN5 by 19 Å (Supplementary Fig. 9e).

**Structure of the deneddylated CSN–CRL2 complex.** Having determined the structure of the CSN–CRL2~N8 complex, we next sought to detail any conformational differences in the deneddylated CSN–CRL2. The affinity of CSN for non-neddylated CRLs is significantly lower than for the neddylated forms, limiting the yield of the desired product[19]. To stabilise the formation of a complex between CSN and CRL2, we employed GraFix[27] ("Methods") prior to cryo-EM. Moreover, native MS confirmed

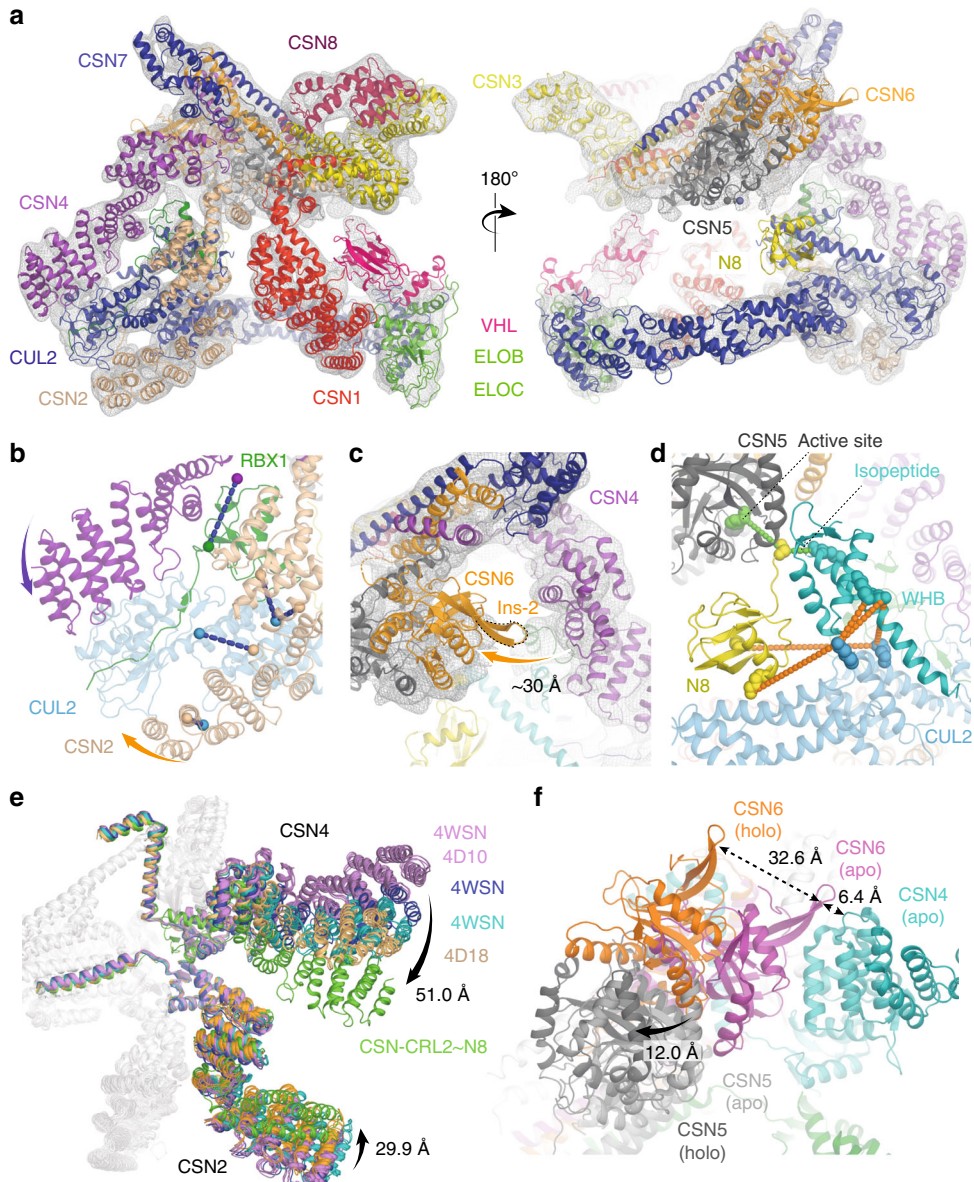

**Fig. 1** Structures and interactions of the CSN–CRL2~N8 complex. **a** The molecular model of CSN–CRL2~N8 fitted into cryo-EM density (8.2 Å resolution) from front and back views. **b** Conformational clamping of CRL2~N8 by CSN2 and CSN4. Cross-links shown are between CSN4-RBX1 (CSN4$^{K200}$-RBX1$^{K105}$, purple-green spheres), and four between CSN2-CUL2 (CSN2$^{K157}$–CUL2$^{K489}$, CSN2$^{K263}$–CUL2$^{K462}$, CSN2$^{K225}$–CUL2$^{K462}$, CSN2$^{K64}$–CUL2$^{K404}$, beige-blue spheres). **c** View showing ~30 Å separation of CSN6 Ins-2 loop from CSN4 following CRL2~N8 binding. **d** Modelled position of WHB~N8 using cross-links of the CSN–CRL2~N8. Large scale conformational changes (**e**) between CSN2 and CSN4 in all apo-CSN crystal structures (PDB 4WSN, 4D10 and 4D18) and CSN-CRL2~N8 (**f**) CSN5/CSN6 (PDB 4D10) upon binding of CRL2~N8 (holo). Subunits of the CSN–CRL2~N8 were compared with the apo-CSN crystal structure (PDB 4D10) following structural alignment. The structure of the CRL2~N8 has been hidden for clarity

the formation of CSN–CRL2 complex (Supplementary Fig. 10, Supplementary Data 1). We next resolved a cryo-EM map of the CSN–CRL2 to 8.8 Å resolution (Supplementary Figs. 11, 12). Although the resolution of the CSN–CRL2 map is similar to that for CSN–CRL2~N8 holocomplex, we only observed partial density for VHL and CSN4. Next using the same procedure as for the neddylated CSN–CRL2~N8 complex, we fitted CSN and CRL2 subunits into the density map of the CSN–CRL2 complex ("Methods" section). We then utilised cross-link guided modelling to establish the position of the WHB which lacked clear density, similar to the neddylated holocomplex (Supplementary Fig. 13; Supplementary Data 3; "Methods" section).

To determine whether the lack of density for the CSN4 may be due to flexibility of the CSN4 N-terminal domain, we carried out

XL-MS for the apo-CSN complex (Supplementary Fig. 14; Supplementary Data 4). A total of 18 cross-links were identified involving CSN4: 12 of which were intra-CSN4 cross-links, 4 between CSN2-CSN4 and 2 between CSN6–CSN4. We measured the distances of these cross-links on the structure of the apo-CSN (PDB 4D10) and on our cryo-EM fitted model of the CSN–CRL2 (which represented the CSN4 in a lowered conformation; Supplementary Fig. 14). Applying a 35 Å distance threshold, we identified cross-links which were exclusively satisfied in each of the two conformations of CSN4 (Supplementary Fig. 14, "Methods"). The presence of exclusively satisfied cross-links indicate that both conformations have been sampled experimentally and suggest that the apo CSN4 can wave between the two conformations represented by the crystal structure and

CRL2-bound structures, even in the absence of the CRL2 substrate.

To evaluate any local changes across the CSN–CRL2~N8 in the absence of NEDD8, we aligned the cryo-EM models of neddylated and denedddylated holocomplexes using the C-terminal helical bundle as a reference point (Supplementary Fig. 15a, b). We systematically compared the conformations of each subunit (Supplementary Fig. 15c–j, Supplementary Movie 1). Compared with its structure in the CSN–CRL2~N8, the N-terminal helices of CSN2 are shifted by ~21 Å towards the CUL2 C-terminal domain (Fig. 2b). This change in CSN2 in the absence of NEDD8, leads to a structural difference in CUL2 which rotates upwards towards the rest of the CSN by 20 Å (Fig. 2c). The position adopted by CUL2 in the denedddylated holocomplex, places ELOB closer to CSN1, forming a CSN1–ELOB interface (Fig. 2d). An interface between CSN1–ELOB can also be seen in the partial structures of CSN–CRL2~N8 missing VHL and CSN5/CSN6 (Supplementary Fig. 2, Supplementary Fig. 16). The dissociation of VHL and CSN5/CSN6 causes the N-terminus of CUL2 to shift downwards, away from the CSN (Supplementary Fig. 16). The plasticity of the CSN results in changes in the C-terminus of CUL2, which is clamped between CSN2 and CSN4, in order to accommodate this shift. The formation of the CSN1–ELOB interface appears to arise as result of this movement in both the denedddylated structure and the incomplete CSN–CRL2~N8 structures. Interactions between substrate adaptor complexes and CSN1 have similarly been reported for the CSN–CRL1~N8[16] and CSN–CRL4A~N8[20]. RBX1 remains clamped between CSN2 and CSN4 (Fig. 2e).

The most striking conformational differences were observed in CSN6 (Fig. 2f, Supplementary Movie 1). In the absence of NEDD8, CSN6 is dramatically shifted away from its position in the neddylated holocomplex by ~40 Å (Fig. 2f). This previously unknown conformation of CSN6 differs from the conformation captured in our neddylated holocomplex, the CSN–CRL1~N8[19] and CSN–CRL4A~N8[20] structures (Supplementary Figs. 17 and 18). We compared the conformation of CSN6 seen in the structures of apo-CSN (PDB 4D10), CSN–CRL1~N8 (EMD-3401), CSN–CRL4A~N8 (EMD-3315), CSN–CRL4A$^{DDB2}$~N8 (EMD-3316) and our neddylated and non-neddylated CSN–CRL2 complexes through systematic structural alignments and measuring their pairwise root mean squared deviation (RMSD) (Supplementary Figs. 17 and 18). The conformation of CSN6 in the non-neddylated CSN–CRL2 complex shows consistently high RMSD (16–25 Å) when compared with CSN6 in any of the other complexes indicating a high degree of conformational difference. CSN6 in the CSN–CRL2 structure has its Ins-2 loop dramatically shifted away from its apo-CSN conformation, pointing upwards to CSN7B (Supplementary Fig. 17, Supplementary Movie 1). We anticipate that this conformation of CSN6 in our model of the non-neddylated CSN–CRL2 is attributed to the lack of NEDD8. Similar to the neddylated holocomplex, no significant changes were identified in CSN3, CSN7B and CSN8 subunits in the CSN–CRL2. Overall, comparison between the CSN–CRL2~N8 and CSN–CRL2 structure reveal significant conformational rearrangements in CSN5/CSN6 and the N-terminal domain of CSN2.

**HDX-MS reveals a stepwise mechanism of CSN activation**. Having determined the structures of the neddylated and denedddylated CSN–CRL2 complexes, we set off to characterise the local dynamics using HDX-MS. HDX-MS provides peptide-level information on the dynamics of proteins through monitoring the exchange events of amide hydrogens for bulk deuterium in the surrounding solution environment[26,28–33]. Here, we performed a set of two differential HDX experiments to

determine the effect of: (a) CRL2~N8 binding to CSN, denoted as Δ(CSN–CRL2~N8 - CSN), and (b) CRL2 binding to CSN, denoted as Δ(CSN$^{WT}$–CRL2 - CSN$^{WT}$) (Fig. 3a, Supplementary Figs. 19 and 20). Regions that exhibit significant HDX differences brought about by the addition of the ligand (i.e. CRL2 and CRL2~N8) are labelled as stabilising (negative ΔHDX; coloured blue) or destabilising (positive ΔHDX; coloured red).

In both Δ(CSN–CRL2~N8 - CSN) and Δ(CSN$^{WT}$–CRL2 - CSN$^{WT}$) experiments, extensive regions in the N-terminal helices of CSN2, CSN4 and the globular domain of RBX1 exhibited stabilisation upon the incubation of CSN with its CRL2~N8 and CRL2 substrates (Fig. 3b; Supplementary Fig. 21a, b). These observations are in line with the conformational clamping by CSN2/CSN4 onto the C-terminal of CRL2 as seen in the cryo-EM structures of neddylated and denedddylated complexes. Within CSN2 of both experiments, we observed significantly destabilised regions around helical modules 6–9. These observations have likely identified the hinge points which permit the bending of CSN2 to clamp onto the CUL2 C-terminus (Fig. 3b; Supplementary Fig. 21a, b). Stabilised peptides belonging to CSN1 (143–156) and ELOB (17–25) were identified in the Δ(CSN$^{WT}$–CRL2 - CSN$^{WT}$) condition, indicating that an interface exists between CSN1–ELOB (Supplementary Fig. 22). It is noted that since the CSN$^{WT}$–CRL2 sample used for HDX were not treated with GraFix (as they were for cryo-EM), it is unlikely that the CSN1–ELOB interface is caused by glutaraldehyde cross-linking. In the neddylated Δ(CSN–CRL2~N8 - CSN) condition, ELOB (17–25) was not found due to the lack of proteomic coverage. However, CSN1 (143–156) is stabilised, suggesting that the CSN1–ELOB interface is present in the neddylated holocomplex.

To assess the affinity between CSN4 and CRL2, we performed PLIMSTEX (protein–ligand interactions by mass spectrometry, titration and HDX) experiments ("Methods"). In these experiments, we performed HDX-MS of CSN or CSN$^{WT}$ in the presence of increasing concentrations of CRL2 or CRL2~N8, to derive dissociation constants (Kd) between CSN4 and CRL2/CRL2~N8. PLIMSTEX differs from differential HDX-MS in several ways. First, no differential comparison is performed from PLIMSTEX. Second, PLIMSTEX experiments utilise a single deuteration time for all samples. PLIMSTEX requires a sufficiently long deuteration time to observe differences between holo and apo states, but short enough to prevent over-deuteration of interfaces as the complex naturally dissociates and reassociates. Finally, PLIMSTEX is a titration experiment which observes deuteration changes as a function of ligand concentration, while differential HDX-MS utilises a 1:1 molar ratio of our CSN and CRL2 substrates.

We tested the affinity of CSN4 for CRL2 or CRL2~N8 in CSN–CRL2~N8, CSN–CRL2 and CSN$^{WT}$–CRL2 complexes. Our PLIMSTEX experiments identified three regions of CSN4 which exhibit a dramatic decrease in deuterium uptake when exposed to the CRL2 substrate. These regions correspond to CSN4 α-helices which are in contact with CRL2~N8 in our cryo-EM fitted model of the CSN–CRL2~N8 (Supplementary Fig. 23). The Kd measurements for the different regions of CSN4 in CSN$^{WT}$–CRL2 (11.3–34.5 nM), CSN–CRL2 (138.1–218.8 nM) and CSN–CRL2~N8 (118.9–389.0 nM) are each in the low nanomolar region, suggesting limited differences in local affinity brought about by changes such as the neddylation status of CRL2. These Kd values for the CSN–CRL2 system all fall within a similar overall range to the published global Kd to the CSN–CRL1 system[19] (1.6–310 nM; Supplementary Table 1), indicating a crucial role for CSN4 in stabilising CSN–CRL2. In addition, HDX-MS based Kd measurements also allowed us to localise individual regions of CSN4 responsible for interacting with CRL2 at the peptide level. (Supplementary Fig. 23).

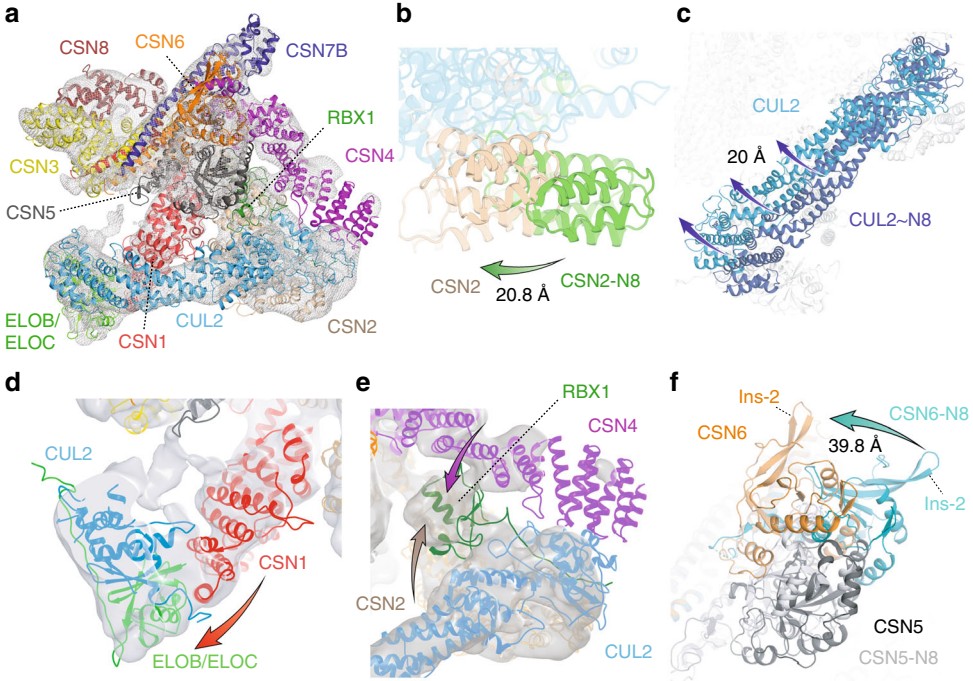

**Fig. 2** Structure of the deneddylated CSN–CRL2 complex. **a** The fitted density map of deneddylated CSN–CRL2 structure determined by combining cryo-EM and XL-MS. Alignment of the CSN C-terminal helical bundle from the neddylated and deneddylated holocomplexes reveals differences in **b** CSN2 and **c** CUL2. **d** CSN1-ELOB interface established from the rotation of CUL2 in **c**. **e** RBX1 is clamped between CSN2 and CSN4. **f** Conformational changes in CSN6 in the absence of NEDD8

**Remodelling of the CSN5 active site in the presence of NEDD8.**
We next considered the release mechanism of the CSN5/CSN6 subunits of both the neddylated and deneddylated holocomplexes. In both $\Delta$(CSN–CRL2~N8 - CSN) and $\Delta$(CSN$^{WT}$–CRL2 - CSN$^{WT}$) experiments, the Ins-2 loop of CSN6 was destabilised, correlating with the release of CSN6 from its interface with CSN4 and in line with the allosteric activation mechanism of CSN by CRL4A~N8[20] (Fig. 4a, b i). An interesting difference between the neddylated and deneddylated complexes is that the CSN6 α4 helix is destabilised only in the absence of NEDD8 (Fig. 4b i). Similarly, the CSN5 α7 helix is also destabilised in both neddylated and deneddylated conditions (Fig. 4a, b ii). The CSN6 α4 and CSN5 α7 helices are topologically knotted in the CSN5/CSN6 heterodimer and tether the globular domains of CSN5/CSN6 to the C-terminal helical bundle[13]. These observations suggest that structural changes are required in the helical knot to bring about release of the CSN5/CSN6 globular domains from their apo conformation.

Another finding is that we identified destabilisation in the Ins-2 loop of CSN5 (Fig. 4a, b ii). The Ins-2 loop of CSN5 has a lesser understood role in CSN activation. In isolated CSN5, the Ins-2 loop is highly disordered[22] (Supplementary Fig. 24a), while it folds into a helical-loop structure when incorporated into the CSN[13] (Supplementary Fig. 24b). Accompanying the changes in the CSN5 Ins-2 loop, in both comparative HDX-MS experiments, we detected destabilisation of the α5 helix area which surrounds the CSN5 active site (Fig. 4a, b ii). The changes in both the CSN5 Ins-2 and α5 helix indicate a major conformational remodelling in the area adjacent to the CSN5 active site, which can be triggered through the binding of both CRL2 or CRL2~N8 to the CSN in a NEDD8-independent manner. It is only in the presence of NEDD8, that the CSN5 active site is further destabilised suggesting that in a final activation step, NEDD8 induces conformational changes in the active site itself (Fig. 4a–d). To eliminate the possibility that the observed changes in the CSN5

active site are due to the H138A point mutation, we compared the deuterium uptake profiles of CSN5 from apo-CSN$^{WT}$ and CSN$^{5H138A}$ constructs (Supplementary Fig. 25). Calculating the deuterium uptake differences between peptides of the CSN5$^{WT}$ and CSN5$^{H138A}$ and visualising this through a Woods plot, identified no significant uptake differences (Supplementary Fig. 25). With this considered, the deprotection observed in the CSN5 active site of the CSN–CRL2~N8 complex can be seen to result from the binding of CRL2~N8 and not the H138A mutation of the CSN5 active site.

**Discussion**
Here we have combined EM and MS analyses to provide insights into the mediation of CRL2 by the CSN. We have described the molecular structures of CSN–CRL2~N8 and its deneddylated CSN–CRL2 counterpart. Furthermore, we combined cryo-EM maps with comparative HDX-MS to expand on the stepwise activation mechanism of the CSN, involving a conformational network of both NEDD8-independent and dependent stages. We suggest that the steps which lead to deneddylation are mostly NEDD8-independent, except for the remodelling of the CSN5 active site which requires NEDD8 to encounter the CSN5 active site.

Our map of the deneddylated CSN–CRL2 holocomplex represents a complex in which the CSN is still associated with its CRL2 reaction product. Resolving this structure has provided several important details into how activation of the CSN is achieved. Our comparison of the neddylated and deneddylated holocomplex structures indicated that the CSN2 contacts the CRL2 C-terminal domain in a slightly different conformation to when the CRL2 is modified with NEDD8. Between both neddylated and deneddylated conformations, we suggest that the clamping by CSN2 involves destabilisation of the CSN2 helical modules 6–9 which function possibly as a hinge that allows the

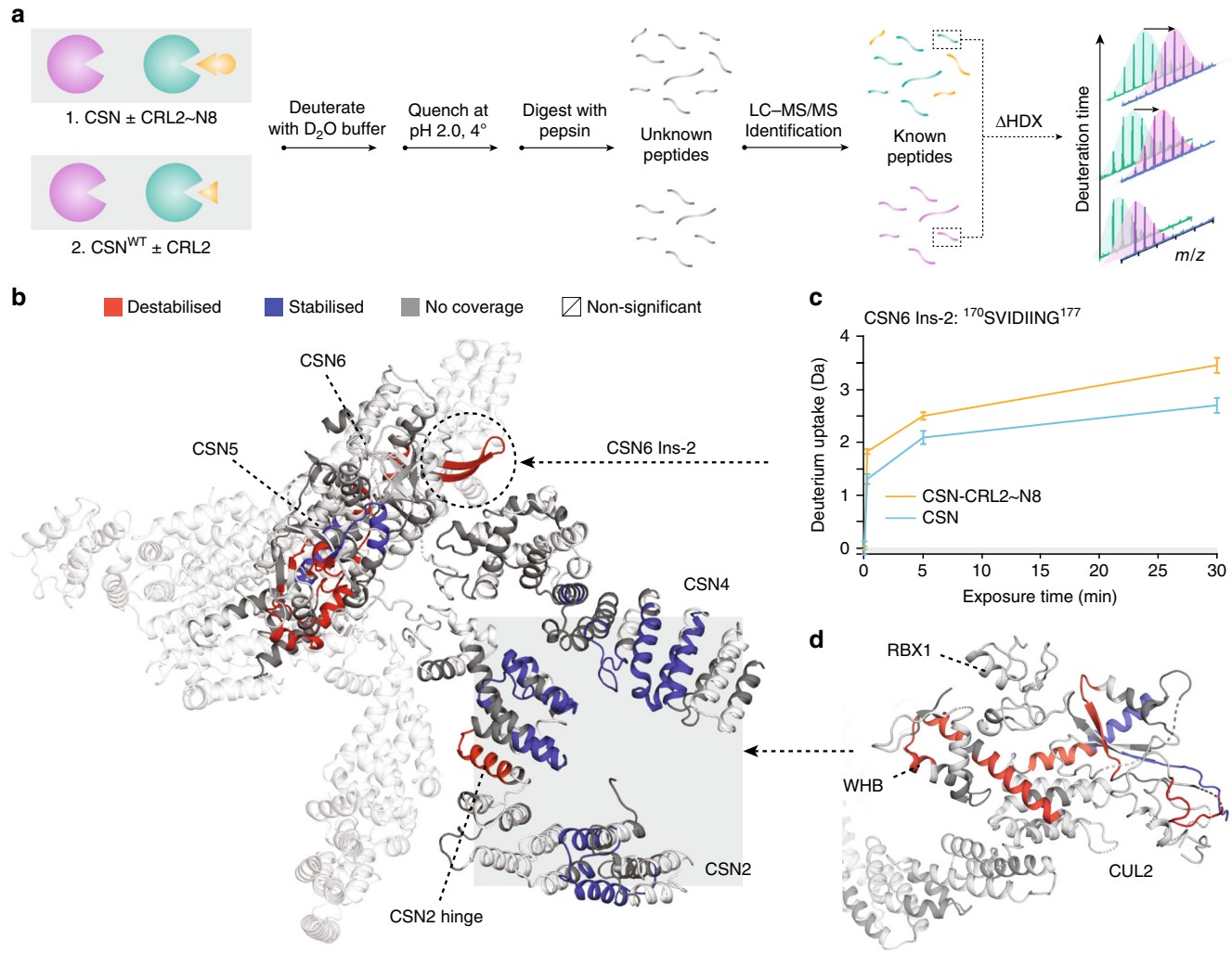

**Fig. 3** Effect of NEDD8 on the CSN4/CSN6 interface. **a** Investigating the effect of CRL2~N8 and CRL2 binding on CSN and CSN[WT], respectively. The two experiments involve deuterating the complexes for 0.25, 5 and 30 min time points, a quench step to halt the deuteration, and digestion to the peptide level. Peptides are then identified using liquid chromatography-tandem mass spectrometry (LC-MS/MS) and a database search. **b** Effect of CRL2-N8 binding on the CSN. **c** Relative deuterium uptake over 30 min for the CSN6 Ins-2 peptide ([170]SVIDIING[177]). Error bars indicate the deuterium uptake standard deviation for triplicate measurements ($N = 3$). **d** Effect of CSN on the C-terminal domain of CRL2-N8. Colour scheme follows that of **b**. Source Data are provided as a Source Data file

CSN2 to bend upwards towards the CRL2. The plasticity of these N-terminal helices presumably permits the binding of deneddylated and alternative Cullin isoforms. HDX-MS further indicates that RBX1 and CSN4 form an interface, which is more prominent in the absence of NEDD8, as shown through stabilisation of the two interfaces (Supplementary Fig. 21b). Overall, the conformational variations seen in the CSN2 N-terminal helical modules (Fig. 2b), the bend of CRL2 (Fig. 2c) and the HDX differences in CSN4/RBX1 (Supplementary Fig. 21b) may therefore be attributable to the seemingly promiscuous affinity that allows CSN2 to bind to each of the different Cullins regardless of neddylation.

We uncovered structural and dynamical aspects of both the neddylated and deneddylated CSN–CRL2 complexes. Beginning our interpretation from valuable studies of the CSN–CRL1~N8 and CSN–CRL4A~N8 systems, we propose several major conformational switches of the CSN which must be activated by the CRL2 substrate to bring about deneddylation. While some of these steps are conserved ubiquitously among other CSN-CRL complexes (e.g. CSN2/CSN4 clamping in CSN–CRL1~N8 and CSN–CRL4A~N8), our study suggests additional steps for the CRL2-bound CSN complex (Fig. 5). In the first activation step,

the CSN and CRL2~N8 associate through major conformational changes in CSN2 and CSN4, which clamp onto the CRL2 (Fig. 5a, b). Our data suggest that the conformational change in CSN4 breaks its interface with CSN6 through the CSN6 Ins-2 loop and with the eventual release of the CSN5/CSN6 heterodimer (Fig. 5c). Removal of the CSN6 Ins-2 loop has been shown in CSN–CRL1~N8 to disrupt the CSN4–CSN6 interface, leading to sustained enzymatic activity of the CSN[13]. In future studies, targeted deletion of the CSN6 Ins-2 loop can be performed alongside differential HDX-MS for the CSN–CRL2 system, to further probe the differences in active site remodelling of CSN5 as a result of disrupting the CSN4–CSN6 interface.

The release of CSN5/CSN6 appears consistent with the destabilised knotted helices of CSN6 that our HDX has identified. It is plausible that these two helices function as the mechanical hinges which allow the CSN5/CSN6 to be released from their auto-inhibited conformations but remain tethered to the rest of the CSN. Although the resolution presented by CSN5 in our cryo-EM structures prevents us from making molecular level observations, our HDX data can provide local detail for the CSN5 active site. The release of CSN5/CSN6 is accompanied by HDX changes in

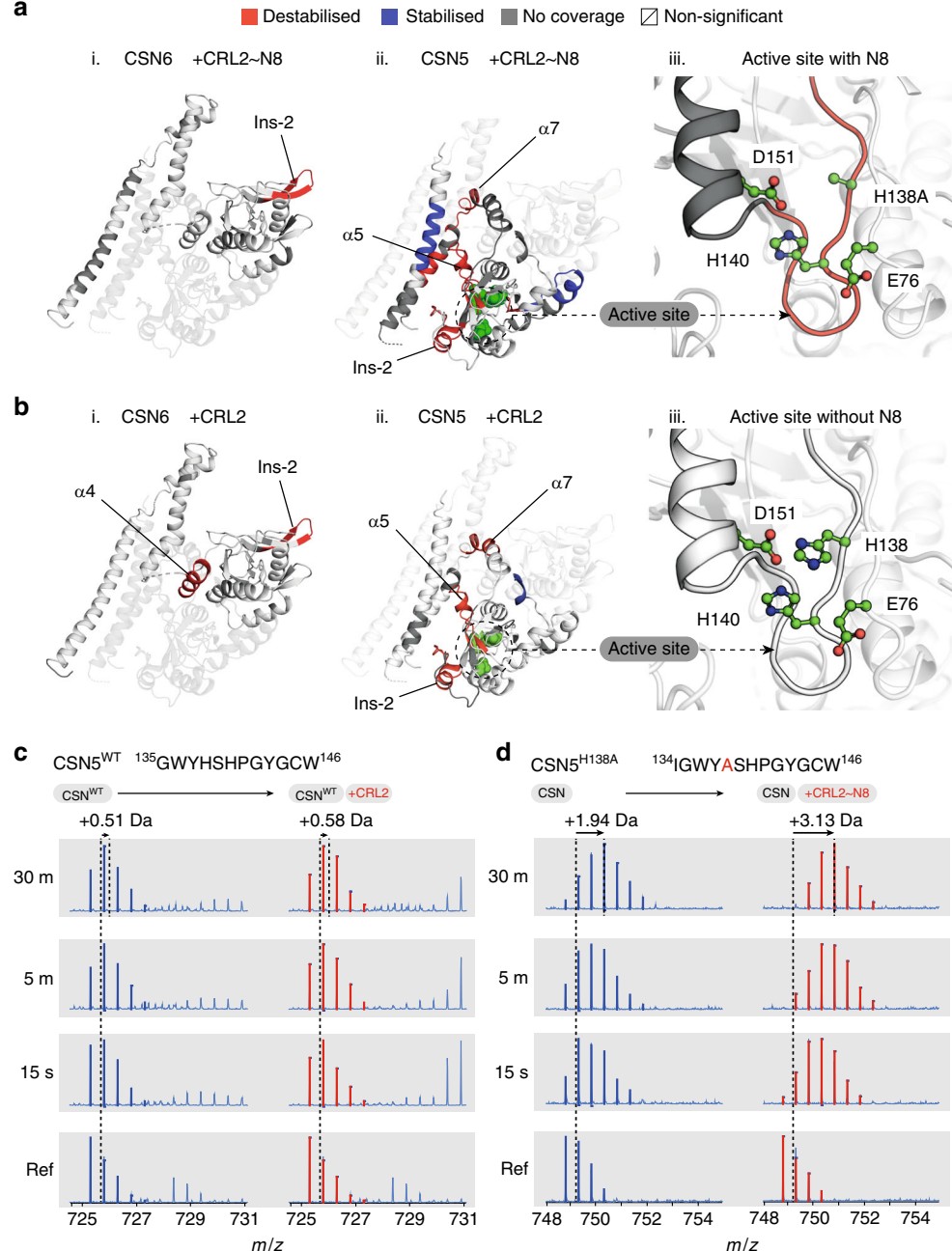

**Fig. 4** Conformational response of CSN5/CSN6 to NEDD8. Differential HDX-MS of **a** Δ(CSN–CRL2-N8 - CSN) and **b** Δ(CSN$^{WT}$–CRL2 - CSN$^{WT}$). Regions exhibiting significant deuterium uptake differences in (i) CSN6, (ii) CSN5 and (iii) CSN5 active site are highlighted in red for destabilised and blue for stabilised areas. Regions without coverage or exhibit non-significant changes are in grey and white, respectively. **c–d** Deuteration profiles of CSN5 active site (as shown in **a–b** iii). Profiles of the **c** non-neddylated and **d** neddylated holocomplex (red) are compared with deuteration profiles of the isolated CSN (blue). The ion distribution of active site peptide is shown for **c** and **d** across reference 0 s, 15 s, 5 m and 30 m timepoints. The mass of the non-deuterated reference peptide is shown by the dotted line. A second dotted line for the holocomplexes (red) indicates the mass of the peptide at the 30 m timepoint. The relative deuterium changes of the active site peptide in **c** following binding of CRL2 is negligible, while in **d**, presence of CRL2-N8 leads to a significant increase in mass. Source Data are provided as a Source Data file

areas surrounding the CSN5 active site, including the CSN5 Ins-2 loop. Up to here, the changes experienced by the CSN can be brought about in a NEDD8-independent manner. In the next stage, the presence of NEDD8 acts as a selectivity filter which results in remodelling of the CSN5 active site itself (Fig. 5d). These changes presumably expose the CSN5 JAMM ligands of the metalloprotease site and allow subsequent deneddylation to occur (Fig. 5e). Finally, deneddylation ensues with the cleavage of NEDD8 from CRL2 and the dissociation of the complex (Fig. 5f).

The fact that CSN can then reassociate with its CRL2 reaction product following dissociation, as shown by our study and structure of the CSN–CRL2, suggests that the non-neddylated complex may possess an alternative role to deneddylation. By associating with non-neddylated CRLs, the CSN sterically blocks access of both ubiquitination E2 enzymes and substrates to the CRLs[16,20]. Further studies will be required to fully understand the deneddylation-independent roles of the CSN. Interestingly, a comparison of our non-neddylated CSN–CRL2 with the apo-CSN

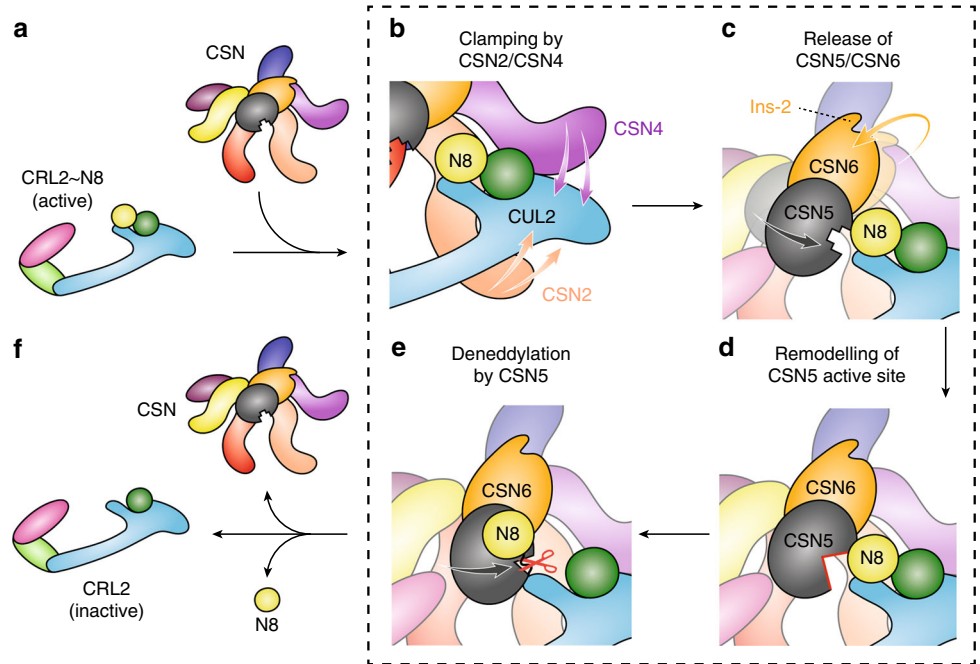

**Fig. 5** Schematic of CRL2 regulation by the CSN. **a** Neddylated (active) CRL2~N8 is regulated by the CSN. **b** CRL2-N8 is bound by the CSN, principally through clamping by the extended N-terminal helical modules of subunits CSN2 and CSN4. **c** The topological changes that occur in CSN4 upon CRL2 binding result in detachment of the CSN6 Ins-2 loop and subsequent release of the CSN5/CSN6 heterodimer in a NEDD8-independent step. **d** The presence of NEDD8 triggers remodelling of the CSN5 active site, **e** permitting deneddylation of CRL2. **f** Cleavage of NEDD8 results in dissociation of the CSN and inactive CRL2

and neddylated CSN–CRL4~N8 and CSN–CRL2~N8 complexes, highlighted the dramatic differences in the conformation of CSN6. In the absence of NEDD8, the CSN5/CSN6 are released much further than any other observed conformation of CSN6. We hypothesise that this difference may arise from the lack of steric hinderance usually presented by NEDD8, allowing CSN5/CSN6 to rotate and meet the Cullin scaffold much closer than in the neddylated structure.

In our study we have made interpretations of the CSN2 and CSN4 conformational clamping through comparing our cryo-EM holocomplexes with crystal structures of the apo-CSN. From reviewing the conformations of all nine currently available independent copies of the apo-CSN molecule it is apparent that CSN2 and CSN4 appear to exist in a range of open conformations that are quite distinct from the closed conformations that we observed in CSN–CRL2 complexes (Supplementary Fig. 8), and the similarly closed conformations observed in CSN complexes with CRL1~N8 and CRL4A~N8[19,20]. However, each of the apo-CSN crystal structures is characterised by crystal contacts involving CSN2 and CSN4 which in principle could have biased their conformations.

Therefore, we used XL-MS to further monitor the conformation of CSN4 within the solution structure of apo-CSN and found that both open and closed conformations of apo-CSN are required to satisfy a number of unique cross-links (Supplementary Fig. 14). Since our study used lysine cross-links, our distance measurements do not allow for a high level of scrutiny in our modelling due to the uncertainty of lysine side chain rotamers in our models. Nonetheless, given that the violated cross-link distances are well above 40–50 Å for one model and close to 35 Å for the other (Supplementary Fig. 14), we can assume that both open and closed conformations are simultaneously represented within the cross-link ensemble. Thus it seems likely that crystal formation has stabilised CSN conformations already present in solution. Hence, while we cannot completely rule out the possibility that the

crystallographically defined open conformations of apo-CSN are atypical, it seems likely that the solution structure of the apo-CSN corresponds to an ensemble containing a range of open structures including those seen crystallographically as well as some closed or nearly closed structures. In comparison, CSN2 and CSN4 in the cryo-EM structures of CSN–CRL2 complexes adopt consistently closed conformations sandwiching the C-terminal end of CUL2 with rather little apparent conformational variation. It is likely here that conformational variation is limited by interactions formed with CUL2. Thus it would appear that the interaction between CSN and CRL2 involves the transition of CSN from an ensemble of conformers with substantial variation in the distance between CSN2 and CSN4 to a bound complex in which CSN2 and CSN4 clamp onto the C-terminal region of CUL2 which is characterised by greatly reduced conformational variation.

To further explore the nature of such clamped complexes, we additionally performed PLIMSTEX experiments which provided localised affinity values between CSN4 and CUL2 for each combination of the CSN–CRL2. From a functional perspective, our PLIMSTEX experiments also determined that the H138A mutation of CSN5 leads to measurable changes in Kd between CSN4 and CRL2. It is also important to note that the Kd values presented are not representative of the global Kd between CSN and CRL2 complexes but are only local affinities between the CSN4 and CRL2 subunits. PLIMSTEX experiments require that the labelling time of the experiment is carefully selected based on preliminary experiments[34]. While powerful, a concern is that if the labelling time is too long, protein interfaces may become over-deuterated, leading to an underestimated deuterium uptake decrease and a higher Kd value. Likewise, some interfaces may appear invisible if the labelling time is not sufficiently long to allow deuteration to occur.

In both neddylated and non-neddylated CSN–CRL2 complexes, we identified an interface between CSN1 and ELOB. So far, CSN1 has been shown to interact with Skp2 and Fbw7 in

CRL1~N8[16] and DDB1 in CRL4A~N8 holocomplexes[20]. Mutations of the CSN1–DDB1 interface did not affect binding to the CSN nor perturb deneddylation activity[20]. In the absence of Cullin 1 or RBX1, CRL1 substrate receptors do not associate with the CSN1[16]. Although no CSN activator role has been assigned to CSN1-substrate receptor interactions, their presence increases the interface between CSN and their CRL substrates[16] and may stabilise these interactions.

Furthermore, there may be additional roles for the CSN as suggested by the compositional plasticity seen in our CSN–CRL2~N8 classes. In our map of the CSN–CRL2~N8, the Cullin arm was observed to shift downwards away from CSN3 in maps deficient of the VHL substrate receptor (Supplementary Fig. 26), consistent with the coupling of VHL binding to a conformational change in the rest of the CRL2 portion of the protein. This type of behaviour may allow the CRL2 to adapt to individual substrates and substrate receptors which vary in size and geometry as has been suggested with the CRL4A system[20]. In addition, it may reflect changes associated with remodelling of the CRL2 by the dissociation of the VHL and the binding of alternative substrate receptors. In future work we will seek to determine whether the CSN can mediate substrate receptor dissociation.

Overall, our study has provided greater detail into the role of CRL2 and NEDD8 in regulating the activation mechanism of CSN. We propose that the series of mechanistic responses of the CSN that lead up to deneddylation, can be triggered even by the CRL2 reaction product in a NEDD8-independent manner. The presence of NEDD8 on the activated CRL2 substrate would then trigger the remodelling within the catalytic site of the CSN5 subunit during the final stage of CSN activation. We envision that this type of mechanism would have implications for the entire family of CRL proteins and their regulatory relationship with the CSN. Our study therefore provides a template not only for assisting investigations of other CRL-based systems but also for bringing together data from different structural biology techniques that otherwise will be reported independently.

## Methods

**Preparation and expression of bacmids**. WT and catalytically reduced CSN[5H138A] bacmids were a kind gift from Radoslav Enchev (The Francis Crick Institute, London)[19]. pcDNA3-myc3-CUL2 was a gift from Yue Xiong (Addgene plasmid #19892[35]). HA-VHL wt-pBabe-puro was a gift from William Kaelin (Addgene plasmid #19234[36]). RBX1, ELOB and ELOC were cloned from cDNA from the Mammalian Gene Collection purchased from Dharmacon. To improve protein yield, an N-terminally truncated (1–53) natural isoform of VHL was also produced for use with MS. Both isoforms of VHL were subcloned into a pET-52b (+) vector (Novagen) to add an N-terminal Step-Tag II. Genes were assembled into pACEBac1 using I-CeuI/BstXI restriction sites via the MultiBac system[37]. RBX1 and CUL2 were assembled into one vector and ELOB, Strep II-VHL(ΔN) and ELOC into a second vector. Correct assembly was confirmed by sequencing of entire genes. DH10EmBacY cells were transformed with each assembly and blue/white selection was performed on L-agar plates containing 50 µg ml⁻¹ kanamycin, 7 µg ml⁻¹ gentamycin, 10 µg ml⁻¹ tetracyclin, 100 µg ml⁻¹ Bluo-Gal (Thermo Scientific) and 40 mg ml⁻¹ IPTG. DH10MultiBac bacmid DNA was isolated from single white colonies. Recombinant baculoviruses were generated in Sf9 insect cells (a clonal isolate from Sf21, Life Technologies #11496015) using standard amplification procedures.

**Expression and purification of recombinant CRL2**. High Five Cells (BTI-TN-5B1-4, from embryonic tissue of the cabbage looper, *Trichoplusia ni*, Life Technologies #B85502) were co-infected with bacmids containing RBX1/CUL2 and ELOB/Strep-II VHL(ΔN)/ELOC and incubated at 27 °C and 130 r.p.m. for 72 h. Cells were harvested by centrifugation at 250 × *g* for 10 min at 4 °C before storage at −80 °C. Freeze-thawed pellets were resuspended in 50 mM Tris pH 7.5, 150 mM NaCl, 2 mM DTT containing complete EDTA-free protease inhibitor tablets (Roche) and Benzonase (Sigma-Aldrich). Cells were lysed by sonication and clarified by centrifugation at 25,000 × *g* for 1 h (Beckman JA-20 rotor). Supernatant was bound to a 3 × 5 ml StrepTrap HP columns (GE Healthcare) in tandem, equilibrated with 50 mM Tris pH 7.5, 150 mM NaCl and 2 mM DTT. Protein was eluted by the addition of 2.5 mM *d*-desthiobiotin. The eluted peak fractions were

concentrated to 2 ml and loaded onto a Superdex 200 16/600 (GE Healthcare) size exclusion column equilibrated with 50 mM HEPES pH 7.5, 150 mM NaCl and 1 mM TCEP. All CRL2 and CRL2~N8 samples used throughout were in 50 mM HEPES pH 7.5, 150 mM NaCl and 1 mM TCEP.

**In vitro neddylation of CRL2**. APPBP1-Uba3, UbcH12 and Nedd8-His were purchased from (Enzo Life Sciences). The neddylation reaction was carried out for 10 min at 37 °C with 8 µM CRL2, 350 nM APPBP1-Uba3, 1.8 µM UbcH12 and 50 µM Nedd8 in a reaction buffer (50 mM HEPES pH 7.5 and 150 mM NaCl) supplemented with 1.25 mM ATP and 10 mM MgCl₂. The reaction was quenched with 15 mM DTT and ice prior to loading onto a 1 ml StrepTrap HP column (GE Healthcare). CRL2~N8 was eluted with reaction buffer supplemented with 2.5 mM desthiobiotin and neddylation was confirmed by SDS-PAGE.

**Expression and purification of recombinant CSN**. High Five Cells were co-infected with the bacmids gifted by Radoslav Enchev (The Francis Crick Institute, London) and protein was expressed as described for CRL2, with an additional Ni-affinity step prior to gel filtration to exploit the His6-tag on the CSN5 subunit. For Strep-affinity chromatography 50 mM HEPES pH 7.5, 200 mM NaCl, 2 mM TCEP and 4% glycerol buffer was used, with the addition of 2.5 mM *d*-desthiobiotin for elution. For Ni-affinity using 2× HisTrap HP columns (GE Healthcare) in tandem, the same buffer was used, but protein was eluted by a 0–300 mM Imidazole gradient across 45 ml. For size exclusion chromatography using a Superdex 200 16/600 (GE Healthcare), the column was equilibrated with 50 mM HEPES pH 7.5, 150 mM NaCl, 1 mM TCEP and 2% glycerol. All CSN and CSN[WT] samples used throughout were in 50 mM HEPES pH 7.5, 150 mM NaCl, 1 mM DTT and 1% glycerol.

**Cryo-EM of CSN–CRL2-N8**. The CSN–CRL2~N8 complex was formed by incubation between CRL2~N8 (1.1× molar excess) and CSN at room temperature for 90 min. The preparation (~0.5 MDa) was subjected to size exclusion chromatography using a Superose 6 Increase 3.2/300 column (GE Healthcare), equilibrated in 15 mM HEPES pH 7.5, 100 mM NaCl, 0.5 mM DTT and 1% glycerol to reduce the contribution of apo components. Fractions from the leading edge of the peak were buffer exchanged into 15 mM HEPES pH 7.5 and 100 mM NaCl using PD SpinTrap G-25 columns (GE Healthcare) before initial assessment by negative stain EM. Cryo-grids were prepared using a Vitrobot (FEI). A cryo-EM dataset was collected beamline M02 from Quantifoil grids with an extra carbon layer at the Electron Bio-Imaging Centre (eBIC—Diamond Light Source, UK) on a Titan Krios 300 kV with Gatan K2 detector (M02), Å pix⁻¹ = 1.06. Movies of 25 frames (dose = 1.85 e Å⁻²) were motion corrected in RELION[38] (2.0) using MOTIONCOR2[39] (01-30-2017) and subsequent CTF estimation of micrographs was performed using CTFFIND4[40] (4.1.5) (Supplementary Fig. 1). Auto-picking selected ~317,000 particles from ~3100 micrographs. Particles were subjected to reference free 2D classification to assess data quality and to remove contaminants selected by auto-picking. This process reduced the particle number to ~250,000. Following particle selection through 2D classification, particles were divided into 15 3D classes. Three of these classes (~69,000 particles) were selected for further classification and processing, as described in Supplementary Fig. 2.

**Cryo-EM of CSN–CRL2**. The CSN–CRL2 complex was formed by incubation between CRL2 (1.1× molar excess) and CSN at room temperature for 90 min. Samples were loaded onto a 5–50% glycerol GraFix[27] gradient containing 0–0.2% glutaraldehyde and ultracentrifuged at 86,000 × *g* for 24 h at 4 °C. Gradients were manually fractionated and the resultant aliquots assessed by SDS-PAGE to determine the extent of cross-linking. Fractions were also assessed using negative stain EM in-house. In order to reduce the glycerol content of samples for cryo-EM, fractions containing the desired complex (as determined by negative stain EM) were pooled together and gel filtered into 15 mM HEPES pH 7.5, 100 mM NaCl and 0.5 mM DTT using a Superose 6 Increase 10/300 GL column (GE Healthcare). Fractions were again assessed by negative stain EM before preparing cryo-grids using a Vitrobot (FEI). A cryo-EM dataset was collected at the Electron Bio-Imaging Centre (eBIC—Diamond Light Source, UK) on a Titan Krios 300 kV with Gatan K2 detector (M02), with a sampling of 1.047 Å pix⁻¹. Movies of 85 frames (dose = 1.0 e Å⁻²) were motion corrected in RELION[41] (3.0) using MOTIONCOR2[39] (01-30-2017) and subsequent CTF estimation of micrographs was performed using CTFFIND4[40] (4.1.5). Auto-picking selected ~309,000 particles from ~6800 micrographs. Particles were subjected to reference free 2D classification to assess data quality and to remove contaminants selected by auto-picking. This process reduced the particle number to ~208,000. Following particle selection through 2D classification, particles were divided into six 3D classes first with alignment, then subsequently without alignment using a mask around CSN5/CSN6 in order to perform focused classification on this area. The map that showed the greatest recovery of detail for CSN5/CSN6 (Supplementary Fig. 11) was then subjected to 3D auto refinement and post-processing. Local resolution was estimated using ResMap[42] as part of the RELION wrapper.

**Band-shift assays**. In order to test the activity of the CSN and CSNWT, 3 µg of each complex was separately incubated with 3 µg of CRL2~N8 at 37 °C for 0, 15, 30, 45 and 60 s. Deneddylation was quenched through rapid denaturation by the

addition of NuPAGE lithium dodecyl sulfate sample buffer (Thermo Fisher) and placing on a heat block, pre-heated to 90 °C. Samples were analysed by SDS-PAGE. Deneddylation in samples with the CSN^WT were confirmed by a band shift the gel band corresponding to CUL2, by comparison with CRL2~N8 and CRL2 controls.

**Homology modelling of the CRL2**. Homology modelling of the CRL2 was necessary due to a combination of missing domain structure for the CUL2 Winged-Helix A (WHA) and VHL subunit in the only crystal structure of CRL2 (PDB 5N4W). Homology modelling was performed through two stages: first, generating a CRL2 structure with a correct WHA domain, and second, generating the complete CRL2 intact with the VHL-ELOB-ELOC adaptor complex. In the first stage, we performed structural alignment of the CRL2 (5N4W) and CRL1 (1LDJ) structures in PyMOL (2.0.6). Using MODELLER[43] (9.16), a single model of the CRL2 was generated using the slow molecular refinement option of MODELLER. The model was manually evaluated for correct fold, including the correct positioning of residues (such as the CUL2 K689 NEDD8-acceptor site) already present in the 5N4W crystal structure. In the second stage, we aligned the CRL2 model with the VHL-ELOB-ELOC-CUL2 fragment (4WQO) to generate a template for homology modelling. The isoform 3 of VHL (missing residues 1–53) was used for modelling to maintain consistency with the experimental construct. Again, a single model of the CRL2 (with VHL/ELOB/ELOC) was generated using the slow molecular refinement option of MODELLER. The final model of the CRL2 shows a RMSD of 3.6 Å when compared with the initial crystal structure (5N4W) but includes a complete WHA domain and VHL subunit. The script used can be found in Supplementary Note 1.

**Model fitting of EM maps**. All models were fitted using CSN subunits sourced from the 4D10 crystal structure (chains A–H) and the CRL2 (VHL-ELOB-ELOC) structure generated and described in the Homology Modelling of the CRL2 section. We performed map fitting first by performing rigid body fitting of the CSN and CRL2 subunits to each map in Chimera[44] (1.13.1rc) then using the MDFF[25] (0.5) feature of NAMD[45] for positional refinement. In the rigid body fitting step, elongated subunits such as CSN2, CSN4 and CUL2 were first dissected into smaller rigid bodies to permit better fitting into their densities. Following map fitting of all CSN and CRL2 subunits, we then converted the structures into MDFF-compatible topology files using the protein structure file builder function of VMD[46] (1.9.3). MDFF was performed in two steps: an initial energy minimisation step (scaling factor = 0.3 for 50,000 steps) which coerced each subunit into their map densities, and a second equilibration run (scaling factor = 10 for 200,000 steps) which applied molecular dynamics to produce structurally and energetically realistic structures. Secondary structure, cis-peptide and chirality characteristics of the initial models were calculated and enforced throughout each step to avoid a loss of internal structure for each subunit. For each of the CSN–CRL2~N8 and CSN–CRL2 structures, subunits/domains which lacked clear density were not included to avoid interference with the fitting of other subunits. These were the WHB domain (CUL2 residues 656–745) for all maps, NEDD8 in all NEDD8-including maps, and VHL in the CSN–CRL2 map. The cross-correlation coefficient which calculates the degree of overlap between the cryo-EM map and a simulated map of the same resolution from the atomic model, are reported for each model in Table 1 of the manuscript.

**Native mass spectrometry**. All spectral data were collected using a SYNAPT G2-Si (Waters Corp., Manchester, UK) high-definition mass spectrometer and samples were ionised using a NanoLockSpray™ dual electrospray inlet source (Waters Corporation) run with positive polarity in sensitivity mode. Capillaries were pulled using a Flaming/Brown P-97 micropipette puller (Sutter Instrument) and coated with Au:Pd (80:20) using a sputter coater (Quorum Q150RS). The following mass spectrometer settings were used: capillary voltage 1.60–1.75 kV, sampling cone of 75–150 V, source temperature of 20 °C, desolvation temperature 150 °C and collision energy of 25–75 eV. Gas pressures were: source $9.3 \times 10^{-3}$ mbar, trap $3.3 \times 10^{-2}$ mbar, helium cell 3.4 mbar, drift tube 2.6 mbar, transfer $3.1 \times 10^{-2}$ mbar and time-of-flight $5.7 \times 10^{-7}$ mbar.

A 1:1 ratio of CSN:CRL2~N8 and CSN^WT:CRL2 at a 5–15 μM concentration were pre-incubated for 1 h prior to buffer exchange. Pre-incubated protein samples were buffer exchanged and desalted using Vivaspin 500 (30 kDa MWCO) centrifugal concentrators (Sartorius) into pH 7.5 150 mM ammonium acetate (four wash steps). All spectra were analysed using MassLynx (4.1, Waters Corp.).

**HDX-MS**. HDX-MS experiments were performed on a Synapt G2-Si HDMS coupled to an Acquity UPLC M-Class system with HDX and automation (Waters Corporation, Manchester, UK). Data were collected in positive polarity in sensitivity mode and calibrated using sodium iodide. The following mass spectrometer settings were used: capillary voltage of 3 kV, sampling cone of 100 V, source temperature of 80 °C and desolvation temperature of 150 °C. Acquisition mass range was set to 50–2000 Da. Gas pressures were: source $6.6 \times 10^{-3}$ mbar, trap $2.9 \times 10^{-2}$ mbar, helium cell 4.5 mbar, drift tube 3.1 mbar, transfer $2.8 \times 10^{-2}$ mbar and time-of-flight $8.2 \times 10^{-7}$ mbar.

Protein samples were prepared at a concentration of 7.5 μM. Isotope labelling was initiated by diluting 5 μl of each protein sample into 95 μl of buffer L (10 mM

potassium phosphate in D$_2$O pD 6.6). The protein was incubated at various time points (0.25, 5 and 30 min) and then quenched in ice cold buffer Q (100 mM potassium phosphate, brought to pH 2.3 with formic acid (FA)) before being digested online with a Waters Enzymate BEH pepsin column at 20 °C. The same procedure was used for undeuterated control, with the labelling buffer being replaced by buffer E (10 mM potassium phosphate in H$_2$O pH 7.0). The peptides were trapped on a Waters BEH C18 VanGuard pre-column for 3 min at a flow rate of 200 μl min$^{-1}$ in buffer A (0.1% FA ~pH 2.5) before being applied to a Waters BEH C18 analytical column. Peptides were eluted over 7 min with a linear gradient of buffer B (8–40% gradient of 0.1% FA in acetonitrile) at a flow rate of 40 μl min$^{-1}$ with a runtime of 11 min. All trapping and chromatography was performed at 0.5 °C to minimise back exchange. MS$^E$ data were acquired with a 25–45 eV transfer collision energy ramp for high-energy acquisition of product ions. Leucine Enkephalin (LeuEnk-Sigma) was used as a lock mass for mass accuracy correction and the MS was calibrated with sodium iodide. The online Enzymate pepsin column was washed with pepsin wash (1.5 M Gu-HCl, 4% MeOH and 0.8% FA) recommended by the manufacturer and a blank run using the pepsin wash was performed between each sample to prevent significant peptide carry-over from the pepsin column. Optimised peptide identification and peptide coverage for all samples was performed from undeuterated controls (three–four replicates). All deuterium time points were performed in triplicate on different samples on distinct samples.

Sequence identification was made from MS$^E$ data from the undeuterated samples using the Waters ProteinLynx Global Server 2.5.1 (PLGS). Processing parameters of PLGS were set to: lock mass for charge 1 of 556.2771 Da e$^{-1}$, lock mass window of 0.4 Da, low energy threshold of 135.0 counts, elevated energy threshold of 30.0 counts and intensity threshold of 750 counts. Workflow parameters were: peptide and fragment mass tolerance set to automatic, minimum fragment ion matches per peptide set to 1, minimum fragment ion matches per protein set to 7, minimum peptide matches per protein set to 3, primary digest reagent set to non-specific, number of missed cleavages 0, false discovery rate of 100%.

The output peptides were filtered using DynamX (3.0) using the following filtering parameters: minimum intensity of 2500, minimum and maximum peptide sequence length of 5 and 30, respectively, minimum MS/MS products of 3, minimum products per amino acid of 0.1, and a minimum peptide score of 5. In addition, all the spectra were visually examined and only those with high signal to noise ratios were used for HDX-MS analysis. The amount of relative deuterium uptake for each peptide was determined using DynamX (3.0) and are not corrected for back exchange. State (listing the deuterium uptake per-peptide, per-timepoint and per-experimental state) and difference (listing the difference in deuterium uptake between identical peptides of two states compared for each state), were exported from DynamX. These files were input to Deuteros[47] (1.0.8) which format the differential data into the Woods Plot format. Statistical filtering of peptides is then performed to identify those which exhibit significant uptake differences between the two states. Deuteros applies a blanket confidence interval (specifically, the 98% confidence interval given as 0 ± DU where DU is the Deuterium uptake threshold in Daltons) across all peptides of each timepoint. Significant peptides for each timepoint are then exported into a formatting script which is used to project the filtered data onto a 3D model of the protein (Supplementary Figs. 19 and 20).

**PLIMSTEX for CSN–CRL2 complexes**. In addition to performing traditional time-resolved HDX-MS experiments, we also performed PLIMSTEX (protein–ligand interactions by MS, titrations and HDX) measured on combinations of CSN/CSN^WT and CRL2/CRL2~N8 to derive dissociation constants (Kd). We performed PLIMSTEX for three complexes: (1) CSN–CRL2~N8, (2) CSN–CRL2 and (3) CSN^WT–CRL2 complexes. The final concentration of CSN or CSN^WT was fixed at 250 nM. CRL2 or CRL2~N8 were titrated at either 1:0, 1:0.1, 1:0.5, 1:1, 1:2 or 1:0, 1:0.1, 1:0.5, 1:1, 1:5 molar ratios of CSN:CRL2 (0–1250 nM). The sample setup for each of the three complexes consisted of five undeuterated references of CSN or CSN^WT, three samples of deuterated CSN or CSN^WT, followed by three of each of the above molar ratios of CSN:CRL2. Datasets for CSN–CRL2~N8, CSN–CRL2 or CSN^WT–CRL2 underwent labelling for either 15 sec or 2 min depending on which exposure time yielded better deuterium uptake differences as a function of [CRL2] or [CRL2~N8]. HDX-MS data acquisition and data analysis using PLGS and DynamX were performed as above. Kd values for each complex were derived using a MathCAD worksheet (v14, Parametric Technology Corp., Needham, USA) kindly provided by Michael Gross (Washington University in St. Louis, USA). Briefly, the deuterium uptake of each peptide as a function of increasing [CRL2] or [CRL2~N8] were fitted using a 1:1 binding model, optimising for three parameters: D$_0$ (initial deuterium uptake in the absence of [CRL2] or [CRL2~N8]), ΔD (maximum decrease in deuterium uptake observed) and β (where β is equal to the association constant, $K_a$, for 1:1 binding). The quality of the fit is calculated as the root mean square of the residuals between experimental datapoints and the model values. The PLIMSTEX data fitting process has been documented in much greater detail elsewhere[48–50].

**Sample preparation for XL-MS**. Twenty microlitres of ~20 μM CSN–CRL2~N8, CSN^WT–CRL2 and apo-CSN^WT were each incubated with 1–5 mM BS3 cross-linker for 1 h at 25 °C and 350 r.p.m. in a thermomixer. After cross-linking, complexes were (i) separated by gel electrophoresis (NuPAGE) followed by in-gel

digestion (CSN$^{WT}$–CRL2, CSN$^{WT}$ and CSN–CRL2~N8) or (ii) digested in solution (CSN$^{WT}$–CRL2) and generated peptides were pre-fractionated by gel filtration (CSN–CRL2~N8). Gel electrophoresis was performed using the NuPAGE system according to manufacturer's protocols. In-gel digestion was performed as described before[51]. Digestion in solution was performed in the presence of RapiGest (Waters) according to manufacturer's protocols. For gel filtration, peptides were dissolved in 30% acetonitrile (ACN), 0.1% trifluoroacetic acid and separated on a Superdex Peptide PC 3.2/30 column (GE Healthcare) at a flow rate of 50 µl min$^{-1}$.

**Mass spectrometry for XL-MS**. Peptides were dissolved in 2% ACN, 0.1% FA and separated by nano-flow liquid chromatography (Dionex UltiMate 3000 RSLC, Thermo Scientific; mobile phase A: 0.1% (v/v) FA; mobile phase B: 80% (v/v) ACN, 0.08% (v/v) FA). Peptides were loaded onto a trap column (µ-Pre-column, C18, 100 µm I.D., particle size 5 µm; Thermo Scientific) and separated with a flow rate of 300 nl min$^{-1}$ on an analytical C18 capillary column (Acclaim PepMap 100, C18, 75 µm I.D., particle size 3 µm, 50 cm; Thermo Scientific), with a gradient of 4–90% (v/v) mobile phase B over 66 min. Separated peptides were directly eluted into a Q Exactive Plus hybrid quadrupole-Orbitrap (CRL2 and CRL2~N8) or an Orbitrap Fusion Tribrid Mass Spectrometer (CSN–CRL2~N8) (Thermo Scientific).

Typical mass spectrometric conditions for the Q Exactive Plus were: spray voltage of 1.6–2.1 kV; capillary temperature of 250 °C; normalised collision energy of 30%, activation Q of 0.25. The mass spectrometer was operated in data-dependent mode. Survey full scan MS spectra were acquired in the Orbitrap from 350–1600 or 2000 $m/z$ with a resolution of 70,000 at an automatic gain control (AGC) target of $3 \times 10^6$. The top 20 most intense ions were selected for Higher Energy Collisional Dissociation (HCD) MS/MS fragmentation in the Orbitrap (isolation window, 1.5 or 1.6 $m/z$). MS/MS spectra were acquired at a resolution of 17,500 at an AGC target of $1 \times 10^5$ or $5 \times 10^4$. Previously selected ions within previous 30 s were dynamically excluded for 30 s. Only ions with charge states 2–7 + were selected. Singly charged ions as well as ions with unrecognised charge state were excluded. Internal calibration of the Orbitrap was performed using the lock mass option (lock mass: $m/z$ 445.120025)[52].

Typical mass spectrometric conditions for the Orbitrap Fusion were: spray voltage of 2.5 kV; capillary temperature of 275 °C; collision energy of 30% and activation Q of 0.25. The mass spectrometer was operated in data-dependent mode. Survey full scan MS spectra were acquired in the Orbitrap from 500–1700 or 2000 $m/z$ with a resolution of 120,000 at an AGC target of $5 \times 10^4$. The most intense ions were selected for HCD MS/MS fragmentation in the Orbitrap (3 s cycle time with a maximum injection time of 128 ms; isolation window, 1.6 $m/z$). MS/MS spectra were acquired at a resolution of 30,000 at an AGC target of $5 \times 10^4$. Previously selected ions within previous 30 s were dynamically excluded for 20 s. Only ions with charge states 3–8+ were selected. Singly and doubly charged ions as well as ions with unrecognised charge state were excluded. Internal calibration of the Orbitrap was performed using the lock mass option (lock mass: m/z 445.120025)[52].

**Data analysis for XL-MS**. Raw files were converted into Mascot generic format (mgf) files using pXtract (http://pfind.ict.ac.cn/software/pXtract/index.html). Mgf's were searched against a reduced database containing CSN and CRL2 proteins using pLink1.0 search engine. Search parameters were: instrument spectra, HCD; enzyme, trypsin; max missed cleavage sites, 3; variable modifications, oxidation (methionine) and carbamidomethylation (cysteine); cross-linker, BS3; min peptide length, 4; max peptide length, 100; min peptide mass, 400 Da; max peptide mass, 10,000 Da; false discovery rate, 1%. Potential cross-linked dipeptides were evaluated by their spectral quality. Circular network plots were generated using the XVis[53] webserver (https://xvis.genzentrum.lmu.de/login.php).

**XL-MS guided placement of the WHB, NEDD8 and VHL subunits**. Cross-links determined from XL-MS for the CSN–CRL2~N8 and CSN–CRL2 complexes were used to clarify the position of the WHB, NEDD8 and VHL subunits which lacked clear density in our cryo-EM maps. We performed XL-guided placement using the Integrative Modelling Platform (IMP)[54] (2.6), using as input, the map-fitted models of the CSN–CRL2~N8 and CSN–CRL2. The CSN–CRL2~N8 model post-map fitting, included all subunits except the WHB and NEDD8. Similarly, the CSN–CRL2 model included all subunits except the WHB and VHL. Separately, the subunits of each complex were initialised as coarse-grained bead models, representing each residue as a single bead. The WHB domain (CUL2 residues 656–745) and VHL was sourced from the homology model of the CRL2 (detailed in the section Homology Modelling of the CRL2). NEDD8 was sourced from the crystal structure of neddylated CRL5 (3DQV). WHB, NEDD8 and VHL were set as mobile rigid bodies, while all other subunits were kept stationary.

Our modelling procedure utilised two types of cross-links. The first type are pseudo-cross-links that maintain the correct topology of the complex: a single pseudo-cross-link between CUL2$^{T655}$ to WHB$^{T656}$ of 5 Å to mimic a covalent bond, and connections between VHL-ELOB, VHL-ELOC and VHL-CUL2 to maintain integrity of the VHL-ELOB-ELOC adaptor complex and its interface with CUL2. A single pseudo-cross-link of 10 Å was used to mimic the isopeptide bond of WHB$^{K689}$~N8$^{G76}$ (7.5 Å lysine side chain + ~3 Å glycine C-terminus). The second type are cross-links determined experimentally between WHB, NEDD8 and VHL with its surrounding subunits (Supplementary Data 2, 3) which utilised a

distance threshold of 35 Å (two lysine side chains at 15 Å, BS3 linker length at 10 Å, plus 10 Å for flexibility). IMP was parametrised to perform 1000 iterations, with each iteration randomly moving WHB, NEDD8 and VHL relative to the stationary CSN and CRL subunits. IMP parameters used were num_mc_steps = 10, rb_max_trans = 2, rb_max_rot = 0.1, bead_max_trans = 0.5 and excluded volume restraint resolution = 20. The single best model was evaluated by projecting all cross-links for the complex onto the structure and confirming that all distances were below the 35 Å distance threshold. A table of cross-links can be found in Supplementary Data 2 for the CSN–CRL2~N8 and Supplementary Data 3 CSN$^{WT}$–CRL2. The script used can be found in Supplementary Note 2.

**Reporting summary**. Further information on research design is available in the Nature Research Reporting Summary linked to this article.

## Data availability

The cryo-EM density maps described here have been deposited in the Electron Microscopy Data Bank (EMDB) under EMD-4739, EMD-4744, EMD-4742, EMD-4736 and EMD-4741. All of the model coordinate sets fitted to the cryo-EM maps have been deposited in the Protein Data Bank (PDB) under 6R7F, 6R7N, 6R7I, 6R6H and 6R7H. All XL-MS, HDX-MS and native MS data were deposited to the ProteomeXchange repository under the accession codes PXD013001 and PXD013018. A list of data collected for each technique and complex is available in the Supplementary Data 1. The source data for Figs. 3c, 4c, d and Supplementary Figs. 2, 3, 6, 9a, c, 10, 11d, 13a, c, 14c, 18, 19, 20, 22b, 23b and 25 are detailed in the Source Data file. All other data are available from the corresponding authors on reasonable request.

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

## Acknowledgements

We thank Radoslav Enchev (The Francis Crick Institute, London) for the COP9 baculovirus plasmids, and Lucie Vyletova and Ruth Knight (Institute of Cancer Research, London) for their help with baculovirus expression and insect cell maintenance. We are grateful to Chris Richardson (Institute of Cancer Research, London) for IT support and to Jane Sandall (Institute of Cancer Research, London) for laboratory support. We thank the staff at eBIC (Diamond Light Source), particularly Dan Clare and Yuriy Chaban, for their support during cryo-EM data collection. We thank Antoni Borysik (King's College London) for helpful guidance with HDX-MS instrumentation. We thank Jürgen Claesen (Hasselt University, Belgium) for helpful interpretation of the HDX-MS data. We thank Michael L. Gross and Don L. Rempel for providing their PLIMSTEX data analysis software. The London Interdisciplinary Biosciences Consortium (LIDo) BBSRC Doctoral Training Partnership (BB/M009513/1) supports A.M.C.L. S.V.F., E.P.M. and F.B. are funded by Cancer Research UK (C12209/A16749). C.S. acknowledges funding from the Federal Ministry for Education and Research (BMBF, ZIK programme, 03Z22HN22), the European Regional Development Funds (EFRE, ZS/2016/04/78115) and the MLU Halle-Wittenberg. C.M. and A.P. are funded by the Wellcome Trust (109854/Z/15/Z) and a King's Health Partners R&D Challenge Fund through the MRC.

## Author contributions

E.P.M. and A.P. conceived and designed the research. A.M.C.L. and N.B.C. performed molecular modelling; S.V.F., H.Y., F.B. and E.P.M. contributed all protein samples and EM data; Z.A., C.M. and A.P. performed native MS experiments; C.M. and K.H. conducted all HDX-MS experiments; A.M.C.L. conducted all HDX-MS data analysis and interpretation. C.S. performed all XL-MS experiments; A.M.C.L., S.V.F., E.P.M. and A.P. wrote the paper with contribution from all authors.

## Additional information

**Competing interests:** The authors declare no competing interests.

