## [Peer Review File · Nature Communications]

Reviewers' comments:

Reviewer #1 (Remarks to the Author):

Faull et al. have performed a descriptive structural study of CSN-CRL2 complexes using a combination of cryoEM, XL-MS and HDX-MS. They make comparisons between the neddylylated and unneddylylated complexes, as well as with the apo-enzyme determined by Xray crystallography from a prior publication. From their data, the authors have highlighted a number of conformational changes that they observed between the different states of the CSN-CRL2 complexes. They imply that the observed changes are functionally significant, even though no functional follow up has been performed on the structural data to support these implications.

In their cryoEM analysis of the neddylylated holoenzyme, the authors also observe two subcomplexes with missing density of CSN5/6/VHL subunits. It is stated in the manuscript that these partial complexes could arise from compositional heterogeneity. Alternatively, the subunits could still be associated with the complex, but structurally poorly defined and therefore not apparent in the reconstructions. These two possible scenarios can be distinguished with native MS, as it could reveal the compositional heterogeneity. However, no native MS on the neddylylated complex is described in the manuscript.

The authors compare the neddylylated holoenzyme to a published crystal structure of the apoenzyme. They observe large conformational differences between the complexes, most notably of CSN2/4. Considering the different methods used to determine the structures of the holo vs. apo enzymes, have the authors considered that some structural differences between the complexes could be due to crystal contacts? Also, considering the extensive 3D classification performed on the holoenzyme, with their reconstruction representing some 1/10th of the total number of particles in the dataset, have the authors looked whether conformations more like the apoenzyme are also sampled by their holoenzyme? A more direct comparison between the two complexes where the same method is used for both would seem more appropriate.

Next, the authors compare the neddylylated and unneddylylated complex. Even though there is no mention of this in the main text, it appears that only the unneddylylated complex was cross-linked with glutaraldehyde before structural analysis by cryoEM. The two complexes are compared by aligning their fitted models "using the C-terminal helical bundle as a reference point", but it is not clearly defined what constitutes this helical bundle in the text. The most notable conformational changes are observed in CSN2 and CSN6. The conformational change in CSN2 is suggested to introduce a new interface between CSN1 and ELOB. However, given the extensive 3D classification performed for the neddylylated complex it is not clear whether this interaction is not also sampled in

this state. In fact, the 3D classes for the CSN5/6/VHL-lacking complexes in the neddylated sample appear to have CSN1 and ELOB positioned much closer. Moreover, glutaraldehyde cross-linking was performed for only the unneddylated complex, so it is not clear if an interface between CSN1-ELOB should be attributed to cross-linking or neddylation status. A more direct comparison between both glutaraldehyde cross-linked complexes seems more appropriate. Could the new CSN1-ELOB interface in the unneddylated complex not also be verified from the HDX or XL-MS data?

The HDX experiments show several regions that are stabilised or destabilised by the CSN-CRL2 interaction. From the data presented in the manuscript the functional implications of these findings are not entirely clear. Generally speaking, it would be a valuable addition to the manuscript if the authors can verify the functional implications of their structural models with quantitative binding assays, as well as enzymatic assays tracking neddylation/deneddylation activity.

In summary, the data presented is of interest and constitutes an impressive set of hybrid structural methods, but the manuscript generally lacks clear functional validation of the resulting structural models.

Reviewer #2 (Remarks to the Author):

Faull, Lau, and colleagues present a structure-based study on how CSN deactivates CRL by removing NEDD8. Three types of data were collected, EM, XL-MS, and HDX-MS. The EM and XL-MS Data were integrated, and modeling tools were used to produce structures of the complexes. A speculative mechanism was presented in the discussion, based on the structures and the HDX-MS data. The study was well designed, and the data support the claims. I will, therefore, recommend Nature Communications to consider the manuscript for publication.

Concerns:

1. Please add a table detailing the samples that were used in this study and which data was collected on which sample. In particular, was the XL-MS and HDX-MS data collected on the same or distinct samples?
2. Please add a table indicating the file names for all collected data, referencing the sample table in point 1.

3. Please make PDBs, PyMOL sessions or both available for the final models. Include the electron densities, the cross-links, and the protection/deprotection data if possible.
4. The arrows that indicate structural changes are difficult to interpret. I believe a more precise way to display conformational changes would be to include both structures in rainbow color. For example, S9i, it looks like a small part of a helix is moved a bit. I guess that this helix is in loop conformation after the structural change – however, this cannot be read out from the image.
5. Please indicate the exact version used for each software. I cannot find the version of RELION, IMP, Waters MassLynx, and others.
6. Please include the issued commands for each tool. The authors mentioned in-house scripts. Please indicate these in more detail, what they are accomplishing and on what data where were operated. Ideally, make these scripts available.
7. The authors indicate that they will deposit the data, but all data accession codes are placeholders. Please make sure that this data is indeed deposited and made public.
8. In the ‘data availability’ declaration, the authors say that all the other data supporting the findings will be available upon request. I would recommend the authors publishing this data on zenodo.org or similar.
9. If software containers were used, please deposit these and provide links/versions.
10. Why is Deuterios not described in this manuscript? A lot of the plots were generated using this tool, and it is mentioned in the Reporting Summary.

Reviewer #3 (Remarks to the Author):

This study determines the structures of two CSN complexes: one, with CSN bound to a neddylated CRL2; two, with CSN bound to a deneddylated CRL2. The CSN is eight subunit particle with specific isopeptidase activity for cleavage of its substrates, namely the gamut of neddylated CRLs. This biochemical activity places CSN as a master regulator of cellular ubiquitination since CRLs represent a sizable fraction of cellular E3 ligases. Hence, the CSN is a truly important for our understanding of this central pathway, with implications for biomedical research. It has been a target for drug development. In 2014, the x-ray structure of CSN was determined. Two subsequent studies, published in 2016, have characterized its enzymology and structure when bound to neddylated substrates, namely a representative CRL1 and CRL4. No structure has been reported for the isopeptidase-product complex.

The results reported here are then both quite important and novel. They broaden our understanding of enzyme-substrate recognition (here we have a new class of substrate, CRL2). At the same time, they also deepen our grasp of enzymatic mechanism with a snapshot of the isopeptidase bound to product (deneddylated CRL2). The investigators have also introduced two important approaches to buttress the cryo EM findings, that of crosslinking MS and HDX-MS. The former has filled in lacuna arising from the limitations of the EM analysis, allowing them to place dynamic subunits into the quaternary structure. The latter has added a new dimension to the structural characterization: direct information about the dynamic motions of the CSN and its substrate or product. Indeed, perhaps the most elegant insight is the induced fit and “resection” of the CSN5 active site when encountering the neddylated substrate. While the earlier studies had hinted in this direction, the present study provides direct evidence. Finally, the authors posit a mechanistic cycle which can be fodder for future testing.

One feature that could strengthen our understanding here would be biochemical/biophysical assays reporting affinities or enzymatic parameters for the substrate and product. This reviewer appreciates that these assays are technically demanding for this system and may not be readily accessible to the investigators. At the very least, references to any literature for these numbers for the CRL2 would supply the reader with important context. After all, there is more than mechanistic interest in understanding the enzyme-product complex since it has important biological implications for CSN regulation (see below).

Specific comments:

While figure 1 shows the CSN-CRL2~N8 complex, an important result is the conformational changes taken by CSN from apo to holo. This is depicted in figure S4a-b. It seems appropriate to include these panels in figure 1, if possible.

p 9: “Although the resolution of the CSN-CRL2 map is similar to that for CSN-CRL2~N8 holocomplex, we only observed partial density for VHL and CSN4.” Does this mean that VHL and CSN4 dissociate from the complex? Is there any evidence for this dissociation or might it be static disorder?

p 11: In the absence of NEDD8, CSN6 is dramatically shifted away from its position in the neddylated holocomplex by ~40 Å (Fig. 2f). This previously unknown conformation of CSN6 appears to be unique, differing from the conformation captured in our neddylated holocomplex, and the CSN-CRL4A~N8 and CSN-CRL3~N8 structures.” Could you elaborate briefly in the text how precisely this shift is unique i.e. is there a shift at all for those other structures and if so, how is this one different? What might give rise to the difference?

p 13: "It is important to note that our HDX-MS experiments did not cross-compare the CSN5^{H138A} and CSN^{WT} complexes." What does this mean? I take it to be that the neddylated CRL2 was done with the mutant and the deneddylated with WT. If so, then it should be spelled out in the text a couple lines above and in the figure titles and labels (S10, S12, S14). Also, while I fully appreciate that these are differential experiments, nonetheless, there is an important premise left unstated i.e. that the mutant's (CSN5-H138A) active site is not much different than WT so as to disallow the conclusion that WT CSN's active site undergoes remodeling. It would be helpful if the authors could provide arguments for that premise, either by the HDX data or other findings. The necessity of using the mutant is evident. The reader needs to be convinced that figure S14b happens in the WT context. Also, since S14 shows directly the experimental evidence, I think it should be incorporated into figure 4.

p 15: "The CSN6 α 4 and CSN α 7 helices are topologically knotted in the CSN5/CSN6 heterodimer ..." The second CSN mention is missing the number 5.

p 19: "then re-associate with its CRL2 reaction product following dissociation, as shown by our study and structure of the CSN-CRL2, suggests that this complex may serve a lesser understood role in the CSN-CRL network." I guess the authors refer here to the "CSN paradox." [Dubiel, W. Mol Cell, 2009]. It is appropriate to elaborate briefly, since it is quite central to the biology or clarify their point, if that is not the intention.

Supplementary materials

p 9: "following neddylation and finding to CSN." Should be binding to CSN.

p 19: Was there an excess of CRL2 in this experiment or is binding sub-stoichiometric?

p 26: CSN3 should be colored and labeled.

Point-by-point response

Reviewer #1 (Remarks to the Author):

Faull et al. have performed a descriptive structural study of CSN-CRL2 complexes using a combination of cryoEM, XL-MS and HDX-MS. They make comparisons between the neddylated and unneddylated complexes, as well as with the apo-enzyme determined by X-ray crystallography from a prior publication. From their data, the authors have highlighted a number of conformational changes that they observed between the different states of the CSN-CRL2 complexes. They imply that the observed changes are functionally significant, even though no functional follow up has been performed on the structural data to support these implications.

We thank the reviewer for their detailed analysis of our manuscript.

1. In their cryoEM analysis of the neddylated holoenzyme, the authors also observe two subcomplexes with missing density of CSN5/6/VHL subunits. It is stated in the manuscript that these partial complexes could arise from compositional heterogeneity. Alternatively, the subunits could still be associated with the complex, but structurally poorly defined and therefore not apparent in the reconstructions. These two possible scenarios can be distinguished with native MS, as it could reveal the compositional heterogeneity. However, no native MS on the neddylated complex is described in the manuscript.

We performed new native MS experiments in order to better characterise the heterogeneity of the CSN-CRL2 and CSN-CRL2~N8 complexes observed our cryo-EM density maps. We subjected both CSN^{WT}-CRL2 and CSN^{5H138A}-CRL2~N8 to native MS (Figures R1). We observed the dissociation of VHL, ELOB, ELOC and CSN5 from subcomplexes of CSN^{5H138A}-CRL2~N8, consistent with our original interpretation of the heterogeneity in our cryo-EM analysis (**Supplementary Figures 1-2**). These native MS experiments highlight the compositional heterogeneity of CSN-CRL2 complexes due to the stable combinations of subcomplexes that can arise from the pairing of CSN and CRL2. These observations also support the idea that similar levels of heterogeneity observed in the cryo-EM analysis of other CSN-CRL complexes, CSN-CRL1 and CSN-CRL4A (Mosadeghi et al., 2016, *eLife*; Cavadini et al., 2016, *Nature*) is also likely to arise from variable subunit composition. Overall it seems likely that compositional heterogeneity is a characteristic of both CSN and CRLs and may be ubiquitous to all CSN-CRL complexes. We have added these spectra to our supplementary

information as **Supplementary Figures 5 and 9** and included the relevant descriptions in the main text (p. 7).

Rebuttal Figure 1. Native MS of (a) CSN^{WT}-CRL2 and (b) CSN^{SH138A}-CRL2~N8 complexes. Spectra were collected from a 1:1 ratio of CSN^{WT}/CSN^{SH138A} and CRL2/CRL2~N8 following a 1-hour incubation at room temperature. The identity, charge, observed mass and mass difference (compared to expected mass) is shown for each complex. CSN3 and CSN5 subunits included 2x StrepII and 6HIS N-terminal tags, respectively. The mass of all CSN subcomplexes include tag masses of CSN3 and CSN5. The presence of the VBC adaptor complex on CRL2 has been marked explicitly, including any missing subunits from complexes.

2. The authors compare the neddylated holoenzyme to a published crystal structure of the apoenzyme. They observe large conformational differences between the complexes, most notably of CSN2/4. Considering the different methods used to determine the structures of the holo vs. apo enzymes, have the authors considered that some structural differences between the complexes could be due to crystal contacts?

This is a good point and it is indeed possible that crystal contacts might stabilise particular conformations of apo-CSN which are not necessarily the dominant forms in solution. We have therefore addressed this issue by reviewing the conformations of individual copies of apo-CSN and their the crystal contacts involving the N-terminal helical repeats of the CSN2 and CSN4 subunits within the three published CSN crystal structures, 4D10, 4D18 and 4WSN, using the PDBe PISA webtool (<http://www.ebi.ac.uk/pdbe/pisa/>). From these crystal structures a total of 9 distinct copies of the apo-CSN complex can be identified which differ both in their conformation and in the nature and extent of the crystal contacts made by CSN2 and CSN4 with adjacent subunits of neighbouring apo-CSN complexes (**Rebuttal Figure 2**). Individual apo-CSN structures are compared with each other and with those seen in our CSN-CRL2 complexes from cryo-EM in **Rebuttal Figure 2a**. Overall, with respect to CSN2 and CSN4, there are considerably greater conformational variations seen in the apo-CSN crystal structures, compared to the different cryo-EM derived CSN-CRL2 complexes. Importantly, in all of the apo-CSN structures the N-terminal helical repeats of both CSN2 and CSN4 are substantially displaced away from their positions in the CSN-CRL2 complex. The maximal displacement for CSN2 and CSN4 (29.9 Å and 51.0 Å, respectively) is seen in the 4D10 crystal structure as described in our original submission (**Figure 1e**). We have updated **Figure 1e** replacing it with **Rebuttal Figure 2a** to illustrate the range of conformations in the multiple structures of apo-CSN.

To further test whether the conformational variation of CSN4 is also observed in a solution environment, we performed additional cross-linking mass spectrometry experiments on the apo-CSN^{WT} complex using the BS3 cross-linker (targeting amine groups). Overall, we identified 86 cross-links, of which 16 were involved CSN4: 11 were between CSN4-CSN4 (intra-protein) and 5 between CSN2-CSN4 (inter-protein).

We then measured the distances of these cross-links on the 4D10 crystal structure of apo-CSN^{WT} (representing one of the most extended CSN4 conformations) (**Rebuttal Figure 2c**) and our EM-map

fitted model of the CSN-CRL2 (which represented the CSN4 in a “lowered” conformation) (**Rebuttal Figure 2d**). By subsequently applying a 35 Å distance cut-off (accounting for two lysine side chains at 7.5 Å each, 10 Å for the BS3 cross-linker length and an additional 10 Å for local flexibility), we identified cross-links which were exclusively satisfied in each of the two models (**Rebuttal Figure 2e**; indicated with arrows). Since the model of CSN4 in both extended and lowered conformations, exclusively satisfy certain cross-links, this indicates that both conformations are sampled in solution. These data imply that N-terminal domain of CSN4 is flexible in solution and can “wave” between both conformations. This “waving” motion is likely not restricted to the apo-CSN since published EM-density maps of the CSN-CRL1 and CSN-CRL4A, also lack density in the N-terminal region of CSN4, suggesting that this dynamic behaviour is maintained even when the CSN is engaged by CRLs. Considering this observation, we have added this rationale in the main text (**p. 10**) along with **Rebuttal Figure 2c-e** to our supplementary document as **Supplementary Figure 12**. We have also added the apo CSN cross-links in **Supplementary Table 3** of the supplementary document.

c CSN4 cross-links on apo-CSN crystal structure

d CSN4 cross-links on CSN-CRL2 model

Rebuttal Figure 2. Conformations of CSN2 and CSN4 in apo-CSN and CRL-bound structures. (a) CSN2 and CSN4 structures highlighted according to their conformations. Apo-conformations are shown in purple, blue, teal and orange (PDB 4WSN, 4D10, 4D18). CRL1, CRL2 and CRL4 bound conformations of CSN2 and CSN4 are shown in green. (b) Interface analysis between CSN4 and crystal contacts for each apo-CSN crystal structure. All values calculated using PDBe PISA. (c) Cross-links of CSN4 from apo-CSN^{WT}. Location and distances of CSN4 cross-links in apo-CSN^{WT}, projected onto the structures of apo-CSN crystal structure (PDB 4D10) and (d) cryo-EM map fitted model of CSN-CRL2. Cross-links with distances satisfied under the cut-off of 35 Å are shown by dashed red lines, while those that are satisfied are in blue. (e) Bar plot showing distances of cross-links from (c-d). Dashed line represents a 35 Å distance threshold. Arrows mark cross-links which are satisfied in one conformation of the CSN but not the other. For K116-K200, K364-K214, K253-K200 and K415-K200 cross-links we included the shortest distance model in the satisfied category due to the distances being close to 35 Å, while the alternative conformation is much greater than 35 Å. All measurements were from lysine NZ atoms.

3. Also, considering the extensive 3D classification performed on the holoenzyme, with their reconstruction representing some 1/10th of the total number of particles in the dataset, have the authors looked whether conformations more like the apoenzyme are also sampled by their holoenzyme? A more direct comparison between the two complexes where the same method is used for both would seem more appropriate.

During the refinement process, we did identify a class without distinct density for CRL2 (dark purple map, **Supplementary Figure 2 ii**). We attempted to refine the particles within this map using 3D classification using the CSN crystal structure as a starting model, however this was unsuccessful. We believe that this indicates that only a small proportion of particles are in the apo conformation. We believe that the CSN-CRL2 sample used is not an appropriate starting point to produce an apo-CSN structure as its conformation may be influenced by CRL2-bound particles. Furthermore, the flexibility of CSN4 in solution, as indicated by our new XL-MS results (**Rebuttal Figure 2**) and analysis of the other CSN-CRL structures described in point 2, lead us to believe that it would be difficult to recover density for CSN4 in an apo-CSN cryo-EM structure.

4. Next, the authors compare the neddylated and unneddylated complex. Even though there is no mention of this in the main text, it appears that only the unneddylated complex was cross-linked with gluteraldehyde before structural analysis by cryoEM.

We have added a sentence to make clear the use of GraFix on the CSN-CRL2 complex in the main text (**p. 10**). This was done due to difficulties in stabilising the non-neddylated CSN-CRL2 complex

which we experienced in preliminary experiments. We would also like to note that GraFix was used in the preparation of the CSN-CLR4A cryo-EM structure by Cavadini et al.

5. The two complexes are compared by aligning their fitted models "using the C-terminal helical bundle as a reference point", but it is not clearly defined what constitutes this helical bundle in the text. Text should be fixed – a figure might help too.

We have now introduced the C-terminal helical bundle in the introduction (p. 4). We have also amended **Supplementary Figure 13** of our supplementary document to include a subpanel (b) showing the alignment of the C-terminal helical bundle.

6. The conformational change in CSN2 is suggested to introduce a new interface between CSN1 and ELOB. However, given the extensive 3D classification performed for the neddylated complex it is not clear whether this interaction is not also sampled in this state. In fact, the 3D classes for the CSN5/6/VHL-lacking complexes in the neddylated sample appear to have CSN1 and ELOB positioned much closer.

Since our first submission, we have revisited our cryo-EM data and the reviewer is correct in suggesting that the CSN1-ELOB interface is present within the neddylated CSN-CRL2~N8 dataset. However, the CSN1/ELOB interface is only restricted to incomplete CSN-CRL2~N8 complexes, such as those missing CSN5/CSN6 and VHL and not the intact complex. We have amended our main text (p. 11) to clarify the existence of this interface in the neddylated CSN-CRL2~N8 maps, and also highlighted this interface in **Supplementary Figure 14**.

7. Moreover, gluteraldehyde cross-linking was performed for only the unneddylated complex, so it is not clear if an interface between CSN1-ELOB should be attributed to cross-linking or neddylation status. A more direct comparison between both gluteraldehyde cross-linked complexes seems more appropriate. Could the new CSN1-ELOB interface in the unneddylated complex not also be verified from the HDX or XL-MS data?

Following the reviewers suggestion, we have reviewed our data for the CSN-CRL2 complex. From HDX-MS of $\Delta(\text{CSN}^{\text{WT}}\text{-CRL2} - \text{CSN}^{\text{WT}})$, we did identify significantly stabilised peptides in both CSN1 and ELOB which correspond to the exact interface residues for the CSN1-ELOB seen from cryo-EM (**Rebuttal Figure 3**). The CSN^{WT} and CRL2 used for HDX experiments did not employ any chemical

fixative. Although our XL-MS data did not contain any cross-links between CSN1 and ELOB, this may be explained by limited accessibility of the CSN1-ELOB interface to cross-linking due to the proximity of CSN1 and ELOB. **Rebuttal Figure 3** is now included in our supplementary data (**Supplementary Figure 20**) and we have revised the main text (**p. 14**) to explain the independent support for the CSN1-ELOB interface obtained from our HDX experiments. Furthermore, we clarify that since the CSN1-ELOB interface is seen in both HDX-MS and cryo-EM, we are confident that the interface is not induced by our use of glutaraldehyde (GraFix). GraFix differs from traditional cross-linking treatments due to the fact that proteins are exposed to a low concentration gradient of the cross-linking reagent (0-0.2% glutaraldehyde) while simultaneously undergoing ultracentrifugation. This procedure makes it unlikely to yield artificial complexes due to the low concentration of fixative used and size fractionation via centrifugation which discourages recovery of complexes that have been inter-molecularly cross-linked (Stark & Chari, 2016, *Microscopy*).

Rebuttal Figure 3. CSN1-ELOB interface of CSN^{WT}-CRL2 in HDX-MS. (a) Structure of CSN-CRL2 fitted to cryo-EM density. Significantly stabilised peptides from CSN1 and ELOB between its interface is highlighted in blue. (b) Deuterium uptake curves over 30 minutes for CSN1 and ELOB peptides are shown. Error bars represent the deuterium uptake standard deviation.

8. The HDX experiments show several regions that are stabilised or destabilised by the CSN-CRL2 interaction. From the data presented in the manuscript the functional implications of these findings are not entirely clear. Generally speaking, it would be a valuable addition to the manuscript if the authors can verify the functional implications of their structural models with quantitative binding assays, as well as enzymatic assays tracking neddylation/denoddylation activity.

This is a valid point and prompted us to perform HDX-MS “PLIMSTEX” (protein-ligand interactions by mass spectrometry, titrations and HDX) experiments to derive dissociation constant (K_d) values for interactions between combinations of CSN^{WT}, CSN^{5H138A}, CRL2 and CRL2~N8 complexes. Specifically, we carried out new experiments of CSN^{WT} and CSN^{5H138A} with increasing concentrations of CRL2 and CRL2~N8, respectively. We subsequently measured the deuterium uptake of peptides from CSN4, identified in each condition after deuterium exchange for a set duration. K_d values between CSN4 in either the CSN^{WT} or CSN^{5H138A}, and CRL2 or CRL2~N8, were derived using a MathCAD worksheet (v14, Parametric Technology Corp., Needham, USA), kindly provided by Michael Gross (Washington University in St. Louis, USA) who developed the PLIMSTEX method. We have added our PLIMSTEX procedure into the methods section.

Rebuttal Figure 4. Dissociation constants (K_d) between CSN4 and CRL2/CRL2~N8 in CSN-CRL2 complexes. (a) Structure and density map of the intact CSN-CRL2~N8 shown for reference. Three peptides identified as interacting with CRL2/CRL2~N8 in PLIMSTEX experiments have been highlighted in red. An example of a non-interacting peptide for which K_d cannot be derived, is shown in orange. (b) PLIMSTEX curve plots showing the deuterium uptake of CSN peptides from (i) CSN-CRL2~N8, (ii) CSN-CRL2 and (iii) CSN^{WT}-CRL2, as a function of increasing concentrations of either CRL2 or CRL2~N8. Data points represent the average deuterium uptake and error bars indicate standard deviation of technical triplicates. The red curve for interacting peptides was fitted using a 3-parameter 1:1 binding model for 250 nM CSN or CSN^{WT}, titrated with CRL2 or CRL2~N8 from 1:0 to 1:5 molar ratios. $K_d \pm$ values denote the root mean square (RMS) of the residuals of the experimental datapoints.

Rebuttal Table 3. Kd values determined for CSN-CRL1 (Mosadeghi et al., 2016, *eLife*) and CSN-CRL2 complexes

	Kd (nM)			
	CRL1	CRL1~N8	CRL2	CRL2~N8
CSN ^{WT}	310.0	-	42.9	-
CSN ^{5H138A}	10.0	1.6	221.7	366.6

Our HDX experiments identified three regions of CSN4 which exhibit a dramatic decrease in deuterium uptake when exposed to increasing concentrations of CRL2 and CRL2~N8 indicating that the CSN4 peptide is involved in substrate binding. These three regions of CSN4 correspond exactly with individual α -helices of the N-terminal helical repeats seen to interface with CUL2~N8 in our cryo-EM fitted model of CSN-CRL2~N8 (**Rebuttal Figure 4a**, red peptides). The Kd measurements for the different regions of CSN4 in CSN^{WT}-CRL2 (10.6-35.7 nM), CSN^{5H138A}-CRL2 (126.9-163.6 nM) and CSN^{5H138A}-CRL2~N8 (98.9-372.9 nM) are each characterized by a similar range, suggesting limited differences in local affinity brought about by changes such as the H138A mutation of CSN5 and the neddylation status of CRL2. These Kd values for the CSN-CRL2 system all fall within a similar overall range to the published global Kd to the CSN-CRL1 system (1.6 nM to 310 nM; **Rebuttal Table 3**), indicating a crucial role for CSN4 in stabilizing CSN-CRL2 and most likely other CSN-CRL complexes. Nevertheless, the reduced dependence on neddylation and H138A mutation compared to CSN-CRL1 indicate that there may be substantial differences between the way in which CSN interacts with CRL2, compared to CRL1. In addition to addressing these issues the HDX-MS based Kd measurements also allowed us localize individual regions of CSN4 responsible for interacting with CRL2 at the peptide level. Thus the peptide corresponding to residues 128-143 located close to the CSN4-CUL2 interface is shown by HDX-MS not to be directly involved in interaction, which is consistent with the model fitted to our cryo-EM map itself indicating close proximity but no direct interaction (**Rebuttal Figure 4a**, orange peptide). This has now been added as **Supplementary Figure 21** and a new paragraph in the main text (**p. 14**).

Moreover, in order to address the reviewer's request to perform enzymatic assays tracking neddylation/deneddylation activity, of the CSN^{WT} compared to the CSN^{5H138A}, we carried out a band shift assay (**Rebuttal Figure 5**). We incubated the CSN^{WT} and CSN^{5H138A} each with the neddylated CRL2~N8 over 0-60 seconds before inhibiting deneddylation through rapid denaturation. When the CSN^{WT} was used, we observed a shift in the gel band of CUL2~N8 to a lower molecular weight early in the reaction, indicating rapid NEDD8 removal by the CSN^{WT} (**Rebuttal Figure 5b**). When the

CSN^{5H138A} mutant was used, we observed very little or no deneddylation activity (**Rebuttal Figure 5c**). This has now been added as **Supplementary Figure 3** and in the main text (**p. 7**).

Rebuttal Figure 5. Deneddylation activity of CSN^{WT} and CSN^{5H138A}. (a) Bands corresponding to denatured CSN and CRL2 complexes. (b) Incubation of CSN^{WT} with CRL2~N8 and (c) CSN^{5H138A} with CRL2~N8 over time. Proteins were incubated at 37°C and reactions were inhibited through the addition of lithium dodecyl sulphate (LDS) and quickly heating to 90°C using a pre-heated heat block to ensure rapid denaturation. Bands corresponding to CUL2~N8 and CUL2 are highlighted for clarity.

In summary, the data presented is of interest and constitutes an impressive set of hybrid structural methods, but the manuscript generally lacks clear functional validation of the resulting structural models.

We hope our new experiments and responses are sufficient to address the reviewer's comments. We hope the reviewer agrees that our substantial new results have clarified the functional relevance of our work.

Reviewer #2 (Remarks to the Author):

Faull, Lau, and colleagues present a structure-based study on how CSN deactivates CRL by removing NEDD8. Three types of data were collected, EM, XL-MS, and HDX-MS. The EM and XL-MS Data were integrated, and modeling tools were used to produce structures of the complexes. A speculative mechanism was presented in the discussion, based on the structures and the HDX-MS data. The study was well designed, and the data support the claims. I will, therefore, recommend Nature Communications to consider the manuscript for publication.

Concerns:

1. Please add a table detailing the samples that were used in this study and which data was collected on which sample. In particular, was the XL-MS and HDX-MS data collected on the same or distinct samples?

&

2. Please add a table indicating the file names for all collected data, referencing the sample table in point 1.

We have added a new **Supplementary File List** that details all of the data and information requested by the reviewer. We would like to clarify that cryo-EM, XL-MS, HDX-MS and native MS are all techniques in which the sample cannot be recovered for further experimentation. Each data acquisition step was performed on a different sample of CSN^{WT}, CSN^{5H138A}, CRL2 or CRL2~N8, expressed and purified with identical methods. The method of protein expression, purification and each data acquisition performed, has been detailed in our methods section.

3. Please make PDBs, PyMOL sessions or both available for the final models. Include the electron densities, the cross-links, and the protection/deprotection data if possible.

All PDB and EMDB codes have been included in **Table 1** of the manuscript. A list of all cross-links are provided as **Supplementary Tables 1-3** in the Supplementary Document.

HDX-MS data have been deposited to ProteomeXchange under accession number **PXD013001**:

<http://proteomecentral.proteomexchange.org/cgi/GetDataset?ID=PXD013001>

The following username and password will be required to access this data:

Username: reviewer80814@ebi.ac.uk

Password: OsBpP4iE

XL-MS and Native MS data have been deposited to ProteomeXchange under accession number **PXD013018**: <http://proteomecentral.proteomexchange.org/cgi/GetDataset?ID=PXD013018>

The following username and password will be required to access this data:

Username: reviewer57032@ebi.ac.uk

Password: Sp2Jl1yY

4. The arrows that indicate structural changes are difficult to interpret. I believe a more precise way to display conformational changes would be to include both structures in rainbow color. For example, S9i, it looks like a small part of a helix is moved a bit. I guess that this helix is in loop conformation after the structural change – however, this cannot be read out from the image.

We have amended **Supplementary Figure 11** (now **Supplementary Figure 13**) to show the per-residue root mean squared distance (RMSD) between the CSN-CRL2 and CSN-CRL2~N8 complexes and have represented the structural differences on the structure using a colour gradient.

5. Please indicate the exact version used for each software. I cannot find the version of RELION, IMP, Waters MassLynx, and others.

We have amended this and added all software versions to the methods section of the manuscript.

6. Please include the issued commands for each tool. The authors mentioned in-house scripts. Please indicate these in more detail, what they are accomplishing and on what data where were operated. Ideally, make these scripts available.

The in-house scripts mentioned here refer to Deuterios, which was published shortly after submission of this manuscript. We have now rewritten this section to properly describe the use of Deuterios in our work (p. 29).

The publication of Deuterios can be found here: <https://doi.org/10.1093/bioinformatics/btz022>

Github repository to code: <https://github.com/andym lau/Deuterios>

7. The authors indicate that they will deposit the data, but all data accession codes are placeholders. Please make sure that this data is indeed deposited and made public.

We believe that this has been addressed in point 3 above. All coordinates and density files have been deposited to the RCSB PDB and EMDB databases. A list of PDB and EMDB codes can be found in Table 1 of the manuscript.

8. In the 'data availability' declaration, the authors say that all the other data supporting the findings will be available upon request. I would recommend the authors publishing this data on zenodo.org or similar.

We have deposited our coordinate files and density maps from cryo-EM, XL-MS, HDX-MS and native MS data to appropriate repositories. Please see the above points 1-3 and our Supplementary File List for the full list of data files and deposition IDs.

9. If software containers were used, please deposit these and provide links/versions.

We did not use any software containers for this project. All software used is listed in the methods along with appropriate references.

10. Why is Deuterios not described in this manuscript? A lot of the plots were generated using this tool, and it is mentioned in the Reporting Summary.

Please see above point 6.

Reviewer #3 (Remarks to the Author):

This study determines the structures of two CSN complexes: one, with CSN bound to a neddylated CRL2; two, with CSN bound to a deneddylated CRL2. The CSN is eight subunit particle with specific isopeptidase activity for cleavage of its substrates, namely the gamut of neddylated CRLs. This biochemical activity places CSN as a master regulator of cellular ubiquitination since CRLs represent a sizable fraction of cellular E3 ligases. Hence, the CSN is a truly important for our understanding of this central pathway, with implications for biomedical research. It has been a target for drug development. In 2014, the x-ray structure of CSN was determined. Two subsequent studies, published in 2016, have characterized its enzymology and structure when bound to neddylated substrates, namely a representative CRL1 and CRL4. No structure has been reported for the isopeptidase-product complex.

The results reported here are then both quite important and novel. They broaden our understanding of enzyme-substrate recognition (here we have a new class of substrate, CRL2). At the same time, they also deepen our grasp of enzymatic mechanism with a snapshot of the isopeptidase bound to product (deneddylated CRL2). The investigators have also introduced two important approaches to buttress the cryo EM findings, that of crosslinking MS and HDX-MS. The former has filled in lacuna arising from the limitations of the EM analysis, allowing them to place dynamic subunits into the quaternary structure. The latter has added a new dimension to the structural characterization: direct information about the dynamic motions of the CSN and its substrate or product. Indeed, perhaps the most elegant insight is the induced fit and “resection” of the CSN5 active site when encountering the neddylated substrate. While the earlier studies had hinted in this direction, the present study provides direct evidence. Finally, the authors posit a mechanistic cycle which can be fodder for future testing.

1. One feature that could strengthen our understanding here would be biochemical/biophysical assays reporting affinities or enzymatic parameters for the substrate and product. This reviewer appreciates that these assays are technically demanding for this system and may not be readily accessible to the investigators. At the very least, references to any literature for these numbers for the CRL2 would supply the reader with important context. After all, there is more than mechanistic interest in understanding the enzyme-product complex since it has important biological implications for CSN regulation (see below).

In response to both reviewer 1 and 3 who suggested meaningful biochemical/biophysical assays to clarify the functional relevance of our study, we have performed a series of additional experiments. Please see the response to reviewer 1, point 8 regarding our K_d measurements of the CSN-CRL2 complexes and deneddylation assays.

Specific comments:

2. While figure 1 shows the CSN-CRL2~N8 complex, an important result is the conformational changes taken by CSN from apo to holo. This is depicted in Supplementary Figure 4a-b. It seems appropriate to include these panels in figure 1, if possible.

We have now amended our **Figure 1**.

3. p 9: "Although the resolution of the CSN-CRL2 map is similar to that for CSN-CRL2~N8 holocomplex, we only observed partial density for VHL and CSN4." Does this mean that VHL and CSN4 dissociate from the complex? Is there any evidence for this dissociation or might it be static disorder?

In our response to reviewer one, we performed additional native MS to characterise our samples (**Rebuttal Figure 1**, Reviewer 1 Point 1). We observed dissociation of VHL but not CSN4, indicating that partial density for VHL is likely due to dissociation. For CSN4 however, we can rule out the possibility that the lack of density for the N-terminal domain of CSN4 in CSN-CRL2 EM map (**Figure 2e** of manuscript), results from CSN4 dissociation, since the rest of the CSN4 can be seen in the EM map.

4. p 11: In the absence of NEDD8, CSN6 is dramatically shifted away from its position in the neddylated holocomplex by ~40 Å (Figure 2f). This previously unknown conformation of CSN6 appears to be unique, differing from the conformation captured in our neddylated holocomplex, and the CSN-CRL4A~N8 and CSN-CRL3~N8 structures." Could you elaborate briefly in the text how precisely this shift is unique i.e. is there a shift at all for those other structures and if so, how is this one different? What might give rise to the difference?

In response to the reviewer, we have performed additional comparisons of CSN6 in our CSN-CRL2 against other available structures:

- Apo CSN (PDB 4D10)
- CSN-CRL1~N8 (EMD-3401)
- CSN-CRL4A~N8 (EMD-3315)
- CSN-CRL4^{DB2}~N8 (EMD-3316)
- CSN-CRL2~N8 (our manuscript)
- CSN-CRL2~N8 (no VHL; our manuscript)
- CSN-CRL2~N8 (VHL refined; our manuscript)

We performed systematic structural alignments between our model of the CSN-CRL2 to each of the above complexes and calculated the RMSD between specifically the CSN6 subunits (**Rebuttal Figure 5-6**). Systematic structural alignments reveal that the Ins-2 loop of CSN6 in CSN-CRL2 is dramatically different from CSN6 in other complexes (**Rebuttal Figure 5**). We anticipate that this novel conformation of CSN6 in our model of the non-neddylated CSN-CRL2 is attributed to the lack of NEDD8. The CSN-CRL2 is the end product of the deneddylation step and was previously undefined until now. We have added these results to our supplementary document as **Supplementary Figure 15-16** to illustrate the differences and the uniqueness of the CSN6 conformation in our model of the CSN-CRL2.

While our original text mentioned a comparison of CSN6 in CSN-CRL3, this was done qualitatively. We did not include the CSN-CRL3 structure in our new analysis due to its low resolution of 27 Å and therefore unreliable density for accurately fitting the CSN6 subunit. We have amended the main text (**p. 12**), removed the comparison to CSN-CRL3 and included a new description of our systematic analysis.

Rebuttal Figure 5. Comparison of CSN6 conformations in published CSN and CSN-CRL complexes. Alignment of CSN-CRL2 (orange) with (a) crystal structure of apo CSN (PDB 4D10), (b) CSN-CRL1~N8 (EMD-3401), (c-d) fitted coordinates of CSN-CRL4A~N8 (EMD-3315) and CSN-CRL4A^{DDB2}~N8 (EMD-3316), (e-g) fitted coordinates of neddylated CSN-CRL2~N8 intact complex, missing VHL and VHL-refined (all purple). The CSN6 Ins-2 loop for each CSN6 has been highlighted for clarity (CSN6 of CSN-CRL2 with dashed ellipses, CSN6 of all other complexes in solid ellipses). Numerical quantification of similarity between CSN6 conformations is presented in Rebuttal Figure 6.

Rebuttal Figure 6. Pairwise RMSD matrix of CSN6 in CSN and CSN-CRL complexes. Matrix indicates the RMSD between comparisons of CSN6 combinations in the apo CSN, CSN-CRL1~N8 (EMD-3401), CSN-CRL4A~N8 (EMD-3315), CSN-CRL4A^{DDB2}~N8 (EMD-3316), CSN-CRL2~N8, CSN-CRL2~N8 (-VHL), CSN-CRL2~N8 (VHL refined) and non-neddylated CSN-CRL2 complexes. The CSN1, CSN2, CSN3, CSN4, CSN7 and CSN8 subunits were aligned in each comparison and the RMSD was calculated between the non-fitted coordinates of CSN6 in each alignment using PyMOL. Aligned models are shown in Rebuttal Figure 5.

5. p 13: “It is important to note that our HDX-MS experiments did not cross-compare the CSN5^{H138A} and CSN^{WT} complexes.” What does this mean? I take it to be that the neddylated CRL2 was done with the mutant and the deneddylated with WT. If so, then it should be spelled out in the text a couple lines above and in the figure titles and labels (S10, S12, S14).

The reviewer has interpreted the quoted sentence correctly. Following our new HDX-MS comparison of the CSN5 active site in the CSN^{WT} and CSN^{5H138A} constructs (detailed in the next point), we have removed this line from the manuscript and added a final paragraph to the results section, documenting the indifference between CSN5^{WT} and CSN5^{H138A} in the context of deuterium uptake (p. 16).

6. Also, while I fully appreciate that these are differential experiments, nonetheless, there is an important premise left unstated i.e. that the mutant's (CSN5-H138A) active site is not much different than WT so as to disallow the conclusion that WT CSN's active site undergoes remodeling. It would be helpful if the authors could provide arguments for that premise, either by the HDX data or other findings. The necessity of using the mutant is evident. The reader needs to be convinced that **Supplementary Figure 14b** happens in the WT context.

We have performed a cross comparison of CSN5 from the CSN^{WT} and CSN^{5H138A} complexes using differential HDX-MS. Our comparison has identified no significantly different peptides, indicating that the H138A mutation does not affect the structure of the CSN5 subunit (Rebuttal Figure 7). While H138 has been mutated, there are no major conformational differences between the two constructs, such that the deuterium uptake signature of CSN5 remains unchanged. It is highly likely that the differences in HDX seen in our comparisons of the CSN^{5H138A}-CRL2~N8 and CSN^{WT}-CRL2 result as a consequence of the neddylation status of the CRL2 substrate (CRL2 and CRL2~N8). We have added this comparison to our supplementary document as a new **Supplementary Figure 23** and added this clarification into the main document (p. 16 main document).

Rebuttal Figure 7. Woods plot comparing deuterium uptake difference of CSN5 peptides from apo-CSN^{WT} and CSN^{5H138A} complexes. Each CSN5 peptide is represented by a horizontal bar and are coloured according to the significance of its uptake difference. Dotted line represents a 98% confidence interval used to filter peptides for statistical significance. All 102 peptides identified from CSN5 showed non-statistically significant changes in deuterium uptake. The position of the CSN5 H138A mutation has been highlighted by the blue bar. Plots generated using Deuterios (v1.08).

7. Also, since S14 shows directly the experimental evidence, I think it should be incorporated into figure 4.

We agree with the reviewer and have now added this.

8. p 15: "The CSN6 α 4 and CSN α 7 helices are topologically knotted in the CSN5/CSN6 heterodimer ..." The second CSN mention is missing the number 5.

We have now amended this error.

9. p 19: "then re-associate with its CRL2 reaction product following dissociation, as shown by our study and structure of the CSN-CRL2, suggests that this complex may serve a lesser understood role in the CSN-CRL network." I guess the authors refer here to the "CSN paradox." [Dubiel, W. Mol Cell, 2009]. It is appropriate to elaborate briefly, since it is quite central to the biology or clarify their point, if that is not the intention.

We have amended our original text to clarify our meaning.

Supplementary materials

10. p 9: "following neddylation and finding to CSN." Should be binding to CSN

This has now been amended.

11. p 19: Was there an excess of CRL2 in this experiment or is binding sub-stoichiometric?

In order to prepare complexes for cryo-EM, a slight (1.1 x) molar excess of CRL2/CRL2~N8 was used. This excess was designed to saturate the CSN molecules, and the rationale behind using an excess

of CRL2/CRL2~N8 was that this lower molecular component would be separated-out using gel filtration or the GraFix gradient used for the non-neddylated complex.

12. p 26: CSN3 should be colored and labeled.

We have now amended the figure.

Reviewers' comments:

Reviewer #1 (Remarks to the Author):

Following the numbering from the authors' rebuttal:

1. The compositional heterogeneity that was proposed based on 3D classification in cryoEM has now been confirmed with native MS.

2. The authors now include a more extensive analysis of CSN2/4 conformations in published crystal structures of the apo state. This revealed considerable conformational flexibility along the same 'clamping' mode as proposed in their holo model. The authors now identify that indeed CSN2/4 are involved in crystal contacts. Whether the authors now still attribute the greater magnitude of CSN2/4 conformational change in their cryoEM reconstruction to a difference in functional state or to methods artifacts is not entirely clear. I maintain that for a direct comparison between the apo and holo complexes, the methods of structure determination should be matched. Clearly, CSN2/4 conformation is restricted in the context of the crystal, while this is not accounted for in the current comparison. The added cross linking analysis does not appear to be very specific to the clamping motion of CSN2/4, as most cross links are too close to the hinge between CSN2/4 and satisfy the distance constraints in both proposed apo and holo state.

3. It is still not clear to me whether the neddylated holocomplex in the authors' cryoEM experiments also samples CSN2/4 conformations more like the apo state crystal structures. Can't the authors compare CSN2/4 conformations of all 3D classes, covering all (more) of their input particles, in their analysis with the apo state conformations?

4. I maintain that for a direct comparison of the neddylated vs unneddylated complex, both states should have been prepared with GraFix. Currently, the cross linking is not controlled for. It is good to see it mentioned in the main text now.

5. The added clarification of the 'C-terminal helix bundle' is very helpful.

6. and 7. The authors now highlight HDX differences as confirmation of the CSN1-ELOB interface. The differences observed are marginally significant by their global 98% CI standard. It is odd that the 17-25 ELOB peptide is not included in the CSN-CRL2~N8 comparisons.

8. The added PLIMSTEX data do not address this point. They simply reproduce the previous HDX findings across a wider range of concentrations. Moreover, the PLIMSTEX measurements do not seem to include the same concentrations for all three sampled conditions, complicating the comparison. Similarly, the gel-shift assay just confirms what the authors have assumed throughout, namely that the mutation in CSN5 interferes with neddylation activity, it does not do anything to test the model they propose based on their structural data.

My main concern was this: the authors propose a model for CSN-CRL2 activity that includes a shift in CSN6, CSN2/4 clamping of CRL2, strengthening of the CSN1-ELOB interface and active site remodeling of CSN5. While the cryoEM and MS data do support some aspects of this model on a structural level, their functional significance is still unclear. The authors could have tested their model by making targeted mutations in their constructs that interfere with the CSN6 shift, the CSN2/4-CRL2 interface, the CSN1-ELOB interface etc. and see how these change the binding affinities and biological activity of the complex.

Reviewer #2 (Remarks to the Author):

The authors present an improved manuscript and have addressed all of my concerns. I, therefore, recommend that Nature Communications consider this work for publication.

Reviewer #3 (Remarks to the Author):

The authors have addressed the reviewers' comments including several new experiments that strengthen the conclusions. The current manuscript is a significant advance to the field and deserves publication.

I think it would be beneficial if the authors could prepare a morph movie (in Chimera) that illustrates to readers the conformational shifts suggested by the different structures. While I appreciate that

this morph is speculative, it will allow scientists to better visualize the end points since static figures make it more difficult to do so, especially in complicated multi-subunit complex as described here.

Specific edits:

p 8: “CSN6 by ~ 30 Å and leads to a ~ 12 Å shift in CSN5 (Figure 1c, Supplemental Figure 1f).” Should be just Figure 1c, 1f.

p 9: “when associated with CRL2 (Supplementary Figure 4).” I don’t think that this is the right figure to be cited.

p 9, last line: “iso peptide bond of NEDD8 is juxtaposed towards to the CSN5 active site.” The word towards should be deleted.

p 16: Figure 4c-d is never explicitly cited or described.

Reviewer #1 (Remarks to the Author):

Following the numbering from the authors' rebuttal:

1. The compositional heterogeneity that was proposed based on 3D classification in cryoEM has now been confirmed with native MS.

- **We are happy to hear that the referee agrees that we have addressed this point satisfactorily**

2. The authors now include a more extensive analysis of CSN2/4 conformations in published crystal structures of the apo state. This revealed considerable conformational flexibility along the same 'clamping' mode as proposed in their holo model. The authors now identify that indeed CSN2/4 are involved in crystal contacts. Whether the authors now still attribute the greater magnitude of CSN2/4 conformational change in their cryoEM reconstruction to a difference in functional state or to methods artifacts is not entirely clear. I maintain that for a direct comparison between the apo and holo complexes, the methods of structure determination should be matched. Clearly, CSN2/4 conformation is restricted in the context of the crystal, while this is not accounted for in the current comparison. The added cross linking analysis does not appear to be very specific to the clamping motion of CSN2/4, as most cross links are too close to the hinge between CSN2/4 and satisfy the distance constraints in both proposed apo and holo state.

- **We have expanded our comparison between the conformation of CSN2 and CSN4 in the 9 crystallographically determined apo structures of CSN and their conformation in our CSN-CRL2 structures from cryo-EM in both the text (page 8) and a new Supplementary Figure 6. In each of the apo CSN structures both CSN2 and CSN4 make crystal contacts with adjacent molecules and significant differences in the conformation of both CSN2 and CSN4 are observed (Supplementary Figure 6a-j). Nevertheless, in all of the available structures the N-terminal domains of CSN2 and CSN4 are substantially spaced apart and would correspond to open conformations.**
- **In comparison the conformations of CSN2 and CSN4 in each of our 5 3D classes of CSN-CRL2 (Supplementary Figure 6k) show much less variation and in each case the N-terminal domains**

of CSN2 and CSN4 adopt a closed conformation in which they sandwich the C-terminal end of the Cullin-2.

- We clarify that our XL-MS measurements do in fact monitor the clamping motion of CSN2 and CSN4. We have identified at least three CSN2-CSN4 cross-links (Supplementary Figure 12a), that in the extended PDB 4D10 model, not only exceed the 35 Å distance threshold, but measure greater than 40-50 Å, while for the compact model they are within the region of 35 Å. This difference indicates that these cross-links are in positions which can be used to monitor the clamping motion. Thus, their proximity to the hinge regions of CSN2 and CSN4 suggested by reviewer 1 is not in fact a problem. We also clarify that cross-links between regions of CSN2 and CSN4 closer to their N-termini would not be expected to be found since this distance greatly exceeds 35 Å for both apo and holo conformations and would also be sterically obstructed in the presence of the CRL2 substrate. Overall, the XL-MS data suggest that in the apo state of CSN the N-terminal domains of CSN2 and CSN4 are somewhat flexible and can “wave” between a range of conformations including the open conformations seen in crystallographic studies.
- Taken together these observations all point to CSN2 and CSN4 adopting more open conformations in apo-CSN and a closed conformation in the CSN-CRL2 complex. Similar conclusions have been drawn from previous structural studies of the interaction of CSN with CRL1 and CRL4 (Mosadeghi et al, 2016, Cavadini et al 2016). However, we accept reviewer 1’s point that the open conformation of the crystallographic structures of apo-CSN may have been influenced by crystal contacts involving CSN2 and CSN4 which we document in Supplementary Figure 6. Overall, we favour the interpretation that crystal formation has stabilised conformations already present in solution, which is consistent with our XL-MS data. We therefore do not consider that in the context of the current study an independent cryo-EM analysis of the solution conformation of apo-CSN is warranted. Nevertheless, we accept that

we cannot completely rule out the possibility that the open conformations of apo-CSN are atypical and this point has been introduced into an expanded discussion on the open and closed conformations of CSN (page 20-21).

Supplementary Figure 6. Conformations of CSN2 and CSN4 in apo and holo CSN. (a-i) Crystal contacts of CSN2 and CSN4 across nine molecules of apo-CSN in PDBs 4WSN, 4D10 and 4D18. Crystal contacts of CSN2 and CSN4 and neighbouring asymmetric units within 5 Å are shown by the green mesh. **(j)** Overlay of apo-CSN molecules in (a-i) showing range of CSN2 and CSN4

conformations. (k) Overlay of holo-CSN molecules in cryo-EM structures of CSN-CRL1 (dark blue), CSN-CRL2 (teal) and CSN-CRL4 (light blue). (l) Overlay of (j) and (k). CSN2 and CSN4 in apo and holo structures of the CSN represent two distinct clusters of conformations.

3. It is still not clear to me whether the neddylated holocomplex in the authors' cryoEM experiments also samples CSN2/4 conformations more like the apo state crystal structures. Can't the authors compare CSN2/4 conformations of all 3D classes, covering all (more) of their input particles, in their analysis with the apo state conformations?

- **We have reviewed our structural analysis of the neddylated holocomplex using as many particles as possible. There is some variation between individual 3D classes which is illustrated in the fitted models shown in Supplementary Figure 6k, particularly with respect to the N-terminal domains of CSN2 and CSN4. Nevertheless, as noted above in response to point 2, all of these conformations are substantially closed and differ significantly from the open conformations observed in the crystal structures of apo-CSN (Supplementary Figure 6a-i). This is covered on pages 21 and 22.**

4. I maintain that for a direct comparison of the neddylated vs unneddylated complex, both states should have been prepared with GraFix. Currently, the cross linking is not controlled for. It is good to see it mentioned in the main text now.

- **We appreciate the referee's point. However, we have explained the reasons for our original choice of experimental approach and take it that they accept that in the circumstances repeating these experiments is not essential.**

5. The added clarification of the 'C-terminal helix bundle' is very helpful.

- **We are happy that this issue is now resolved.**

6. and 7. The authors now highlight HDX differences as confirmation of the CSN1-ELOB interface. The differences observed are marginally significant by their global 98% CI standard. It is odd that the 17-25 ELOB peptide is not included in the CSN-CRL2~N8 comparisons.

- Residues 17-25 of ELOB had no proteomic coverage in our $\Delta(\text{CSN-CRL2}^{\sim}\text{N8} - \text{CSN})$ condition and hence was missing from Supplementary Figures 17 and 20.
- We have made this clear in the results section of our manuscript.

8. The added PLIMSTEX data do not address this point. They simply reproduce the previous HDX findings across a wider range of concentrations. Moreover, the PLIMSTEX measurements do not seem to include the same concentrations for all three sampled conditions, complicating the comparison. Similarly, the gel-shift assay just confirms what the authors have assumed throughout, namely that the mutation in CSN5 interferes with neddylation activity, it does not do anything to test the model they propose based on their structural data.

- We have expanded and clarified the functional relevance of our new PLIMSTEX data (p 14-15 and p 21). PLIMSTEX does not simply reproduce our previous time-resolved HDX findings, but instead are a completely separate set of experiments that provide localised affinity values for the different CSN-CRL2 complexes. A principle difference is that PLIMSTEX requires short labelling times for accuracy, while these short durations may be unsuitable for identifying significant deuterium uptake differences in differential time-resolved HDX-MS experiments. Thus, many interfaces may not be visible to PLIMSTEX but may be accessible by time-resolved experiments. This is another reason why time-resolved experiment are used to probe conformational dynamics since it uses a dynamic range of timepoints. We have clarified these differences in our manuscript (p 14).
- These experiments are performed over a concentration range and changes in HDX are used to derive the K_d of the interaction to peptide level resolution – these are quantitative measurements that the previous HDX-MS do not provide.
- From a functional perspective, we are able to determine for example, that the CSN5 H138A mutation allosterically affects the interface between CSN4 and CRL2. We are also able to pinpoint local regions of interfaces which do and do not contribute to interactions, and we

believe that this information will be invaluable for developing our understanding of interactions between the CSN and its CRL substrates.

- **We have re-fitted the PLIMSTEX curves using the same range of concentrations. The Kd values remain largely unchanged (Supplementary Figure 21). We hope the reviewer is satisfied with this correction.**

My main concern was this: the authors propose a model for CSN-CRL2 activity that includes a shift in CSN6, CSN2/4 clamping of CRL2, strengthening of the CSN1-ELOB interface and active site remodeling of CSN5. While the cryoEM and MS data do support some aspects of this model on a structural level, their functional significance is still unclear. The authors could have tested their model by making targeted mutations in their constructs that interfere with the CSN6 shift, the CSN2/4-CRL2 interface, the CSN1-ELOB interface etc. and see how these change the binding affinities and biological activity of the complex.

- **We appreciate the reviewer's concern regarding extensively testing the biological activity of the complex. These suggestions however constitute a significant amount of work which exceeds the scope of our manuscript, which primarily targets the structural aspects of the assembly. To reflect this, we have toned down the mechanistic interpretations of our manuscript (in particular in the discussion) and further included the limitations of our approach in probing the functional relevance of the supercomplex. Moreover, we have added a line in the discussion pointing to these experimental suggestions as means for future directions (p 20).**
- **We would like to further clarify that mutations of CSN2 and CSN6 as suggested by the reviewer have been previously shown to identify regions of CSN which are critical in its binding and activation of CRL1 and CRL4A (Lingaraju et al., 2014, *Nature*, Cavadini et al., 2016, *Nature*). This also appears to be the case with CRL2 as evidenced in the HDX-MS experiments included in our manuscript (Figure 4). As a consequence, we consider that these additional experiments while useful would offer limited additional relevant information beyond the current HDX-MS**

experiments. For example, our HDX-MS has already indicated that CSN2 and CSN4 associate with both neddylated and non-neddylated CRL2, and that the CSN4/CSN6 interface is disrupted as a result of binding (Figure 4). This is a recurring theme of CSN activation that has been seen in both CRL1 and CRL4A systems. We have further demonstrated that the resulting CSN^{WT}-CRL2~N8 complex is biologically active in our band-shift assay. Finally, we would like to clarify that we do not propose “strengthening of the CSN1-ELOB interface” occurs as a step leading to CSN activation in our model. To avoid confusing our message, we have added an additional paragraph to our discussion which clarifies the relationship between CSN and Cullin substrate receptor complexes (p 21).

Reviewer #2 (Remarks to the Author):

The authors present an improved manuscript and have addressed all of my concerns. I, therefore, recommend that Nature Communications consider this work for publication.

- **We are glad to hear that the reviewer is satisfied with our improvements to our manuscript**

Reviewer #3 (Remarks to the Author):

The authors have addressed the reviewers' comments including several new experiments that strengthen the conclusions. The current manuscript is a significant advance to the field and deserves publication.

I think it would be beneficial if the authors could prepare a morph movie (in Chimera) that illustrates to readers the conformational shifts suggested by the different structures. While I appreciate that this morph is speculative, it will allow scientists to better visualize the end points since static figures make it more difficult to do so, especially in complicated multi-subunit complex as described here.

- **We have prepared a video illustrating the conformational changes as suggested by the reviewer.**

Specific edits:

p 8: "CSN6 by ~30 Å and leads to a ~12 Å shift in CSN5 (Figure 1c, Supplemental Figure 1f)." Should be just Figure 1c, 1f.

- **This has now been corrected**

p 9: "when associated with CRL2 (Supplementary Figure 4)." I don't think that this is the right figure to be cited.

- **This has now been corrected**

p 9, last line: "iso peptide bond of NEDD8 is juxtaposed towards to the CSN5 active site." The word towards should be deleted.

- **This has now been corrected**

p 16: Figure 4c-d is never explicitly cited or described.

- **We have now corrected this. We thank each of the reviewers for their thorough reviews of our manuscript.**